# Synthesizing evidence for the external cycling of $NO_x$ in high- to low-$NO_x$ atmospheres

Chunxiang Ye [1] ✉, Xianliang Zhou[2,3], Yingjie Zhang [1,4], Youfeng Wang[1], Jianshu Wang [1], Chong Zhang [1], Robert Woodward-Massey[1,5], Christopher Cantrell [6], Roy L. Mauldin III[7,8,9], Teresa Campos[10], Rebecca S. Hornbrook [10], John Ortega[10], Eric C. Apel[10], Julie Haggerty[10], Samuel Hall[10], Kirk Ullmann[10], Andrew Weinheimer[10], Jochen Stutz[11], Thomas Karl [12], James N. Smith [13], Alex Guenther [13] & Shaojie Song[14]

External cycling regenerating nitrogen oxides ($NO_x \equiv NO + NO_2$) from their oxidative reservoir, $NO_z$, is proposed to reshape the temporal–spatial distribution of $NO_x$ and consequently hydroxyl radical (OH), the most important oxidant in the atmosphere. Here we verify the in situ external cycling of $NO_x$ in various environments with nitrous acid (HONO) as an intermediate based on synthesized field evidence collected onboard aircraft platform at daytime. External cycling helps to reconcile stubborn under-estimation on observed ratios of HONO/$NO_2$ and $NO_2$/$NO_z$ by current chemical model schemes and rationalize atypical diurnal concentration profiles of HONO and $NO_2$ lacking noontime valleys specially observed in low-$NO_x$ atmospheres. Perturbation on the budget of HONO and $NO_x$ by external cycling is also found to increase as $NO_x$ concentration decreases. Consequently, model underestimation of OH observations by up to 41% in low $NO_x$ atmospheres is attributed to the omission of external cycling in models.

Nitrogen oxides ($NO_x \equiv NO + NO_2$) and gaseous nitrous acid (HONO) perturb the photochemical cycling of peroxy radicals ($RO_2$ and $HO_2$) and hydroxide radicals (OH)[1–6]. Field observations in high-$NO_x$ atmospheres highlight the primary production of OH (and NO) via HONO photolysis[1,6–10]. Across high- to low-$NO_x$ atmospheres, secondary production of OH via the $HO_2$ plus NO reaction is another major source of OH.

HONO and $NO_x$ are closely coupled in their $NO_x$-HONO internal cycling, referred to as internal cycling in this context (Fig. 1). Specifically, heterogeneous reactions of $NO_2$ on ambient surfaces and

[1]State Key Joint Laboratory of Environmental Simulation and Pollution Control (SKL-ESPC), College of Environmental Sciences and Engineering, Peking University, Beijing, China. [2]Wadsworth Center, New York State Department of Health, Albany, NY, USA. [3]Department of Environmental Health Sciences, State University of New York, Albany, NY, USA. [4]School of Ecology and Nature Conservation, Beijing Forestry University, Beijing, China. [5]Department of Chemistry, University of Leeds, Leeds, UK. [6]Université Paris-est Créteil, LISA (Laboratoire Interuniversitaire des Systèmes Atmosphériques), Paris, France. [7]Center for Atmospheric Particle Studies, Carnegie Mellon University, Pittsburgh, PA, USA. [8]Department of Chemistry, Carnegie Mellon University, Pittsburgh, PA, USA. [9]Department of Atmospheric and Oceanic Sciences, University of Colorado Boulder, Boulder, CO, USA. [10]National Center for Atmospheric Research, Boulder, CO, USA. [11]Department of Atmospheric and Oceanic Sciences, University of California, Los Angeles, CA, USA. [12]Institute for Meteorology and Geophysics, University of Innsbruck, Innsbruck, Austria. [13]Earth System Science, University of California, Irvine, CA, USA. [14]State Environmental Protection Key Laboratory of Urban Ambient Air Particulate Matter Pollution Prevention and Control & Tianjin Key Laboratory of Urban Transport Emission Research, College of Environmental Science and Engineering, Nankai University, Tianjin, China. ✉e-mail: c.ye@pku.edu.cn

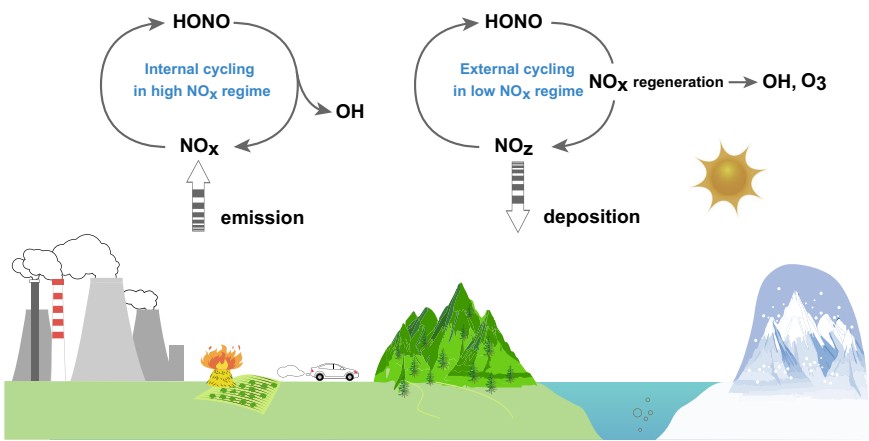

**Fig. 1 | Schematic graph of internal cycling and external cycling of NO and HONO.** The left reaction scheme describes internal cycling between nitrogen oxides (NO = NO + NO) and nitrous acid (HONO) in high- NO atmospheres. HONO photolysis is the major route producing OH radical (OH). The right reaction scheme describes external cycling among oxidative reservoir species of NO (NO), NO, and HONO in low-NO atmospheres. NO -catalysed radical chain propagation takes over as the major route perturbing OH and ozone (O) chemical production.

gas-phase reactions between NO and OH have been intensively examined as major HONO formation routes in the internal cycle[7–15]. Since HONO photolyzes much faster, turnover routes of $NO_x$ to recycle HONO are the rate-limiting steps of internal cycling. OH is a net product in internal cycling even if only HONO production via heterogeneous reactions of $NO_2$ is considered. Any external source (or sink) of HONO or $NO_x$ would promote (or suppress) the internal cycling. Apart from the internal cycling, $NO_x$ ages to form more oxidized reservoirs, which are referred to as $NO_z$, as air masses are transported away from source regions. $NO_x$ aging processes suppress the internal cycling and diminish the role of HONO photolysis in the OH budget. However, model underestimation of the $NO_x/NO_z$ ratio observations suggests unknown or underappreciated $NO_x$ regeneration pathways from $NO_z$ in low-$NO_x$ atmospheres[4,16–28]. $NO_x$ regeneration pathways, in contrast to the internal cycling, produce "new" $NO_x$ and hence are designated as the external cycling of $NO_x$ (Fig. 1). External cycling naturally promotes the internal cycling via at least an external source of $NO_x$. Consequently, secondary OH production via $NO_x$ regeneration and net primary OH production via HONO regeneration could greatly perturb the OH photochemical budget. The current model omission of external cycling would lead to an underestimation of OH production and OH abundance, especially in low-$NO_x$ atmospheres.

Identifying and exploring a specific mechanism is a natural step in characterizing external cycling and quantifying its impact on the oxidative capacity of the atmosphere. The external cycling pathways proposed in the literature include at least three mechanisms, i.e., surface-catalyzed photolysis of absorbed nitrate ($nitrate_{abs}$) on snow/ice surfaces and possibly also on aerosol/ambient surfaces[4,20,24,27–38], nitrification/denitrification in the soil[16,19] and the thermal decomposition of peroxyacetyl nitrate (PAN). Among these mechanisms, surface-catalyzed photolysis of $nitrate_{abs}$ on aerosol surfaces, referred to as $pNO_3$ photolysis, occurs in situ in the air column and therefore potentially perturbs the distribution of $NO_x$ and hence the oxidative capacity of the atmosphere from the lower troposphere to the upper troposphere[35]. $pNO_3$ photolysis has been intensively explored in laboratory and field studies[4,18–23,34,36,37,39]. The reaction rate constant of $pNO_3$ photolysis varies over at least two orders of magnitude in high-to low-$NO_x$ atmospheres by employing a budget analysis for either HONO or $NO_2$ and by assuming that $pNO_3$ photolysis fully accounts for the missing source of HONO or $NO_2$ in the field[4,21,32,34,38]. Laboratory studies on a variety of $pNO_3$ samples have confirmed that $pNO_3$ photolysis is greatly enhanced compared to that of gaseous $HNO_3$ and that the $pNO_3$ photolysis rate constant is highly variable, over 3 orders of magnitude[30,36,37] Based on these laboratory and field

studies, Ye et al. and Andersen et al. have also revealed that $pNO_3$ photolysis is surface-catalyzed in nature and is greatly affected by the physicochemical properties of aerosol particles, such as $pNO_3$ loading, chemical composition and particle size[30,38]. Efforts in characterizing atmospheric aerosol properties and their photochemical reactivities are critical in quantitatively understanding the role of $pNO_3$ photolysis in external cycling. However, the limited availability of the $pNO_3$ photolysis rate constant in only a few atmospheric environments, its large variability, and potentially large uncertainties make it difficult to extrapolate results from laboratory studies to field studies or directly compare results among field studies[18,21,22,30,33,34,36–39]. The variability and potentially large uncertainties in the rate constant have also precluded as yet confident confirmation of $pNO_3$ photolysis or other reactions as the dominant mechanism of the external cycling or direct characterization of the external cycling in the atmosphere. To solve this dilemma, we suggest synthesizing critical observational and model evidence, summarizing the fundamental characteristics, and quantifying the impact of the external cycling on the oxidative capacity of the atmosphere in high- to low-$NO_x$ atmospheres through a broader lens rather than attempting to establish the kinetics or the dominant mechanism of the external cycling.

## Results and discussion
### Synthesized field evidence for external cycling
A comprehensive dataset obtained from an aircraft measurement campaign provided excellent insight into the external cycling of $NO_x$ and its perturbation on the oxidative capacity of the atmosphere[4,32]. Nineteen research flights were conducted onboard NSF/NCAR C-130 to collect measurements of $NO_x$, HONO, nitric acid ($HNO_3$), particulate nitrate ($pNO_3$), some alkyl nitrates, PAN, radicals (OH, $HO_2$, $RO_2$), ozone ($O_3$), volatile organic compounds (VOCs), aerosol size distributions, photolysis frequencies, and other meteorological parameters, mostly in various locations in the low-$NO_x$ troposphere (Methods), i.e., from the pristine terrestrial and marine boundary layer (BL) to the FT (Fig. S1 and Table S1). Herein, we defined low-$NO_x$ and high-$NO_x$ regimes with a $NO_x$ concentration threshold of 500 pptv, given that 500 pptv represented the upper limit concentration of $NO_x$ in the remote troposphere. Furthermore, 500 pptv appeared to be a turning point for the external cycling to be key sources of HONO and $NO_x$ (see below). The data were collected from nineteen research flights without further screening and offered a more global representativity and atmospheric variability of the external cycling and therefore supported our analysis strategy for directly exploring the external cycling of $NO_x$ across high- to low-$NO_x$ atmospheres. In our

previous publications, we exploited data from a limited number of research flights in forested areas (i.e., RF4-5, 11, 17-18) and clean marine boundary layers (i.e., RF14, 16) and discussed the budget of HONO and specific mechanism of the external cycling[4,32]. Several studies of this kind based on aircraft observations had generally employed the budget analysis methodology for HONO to conduct case studies in clean marine air or fire plumes[21,34,38]. The reaction rate constant of the specific external cycling route implied from the missing HONO source among these reports deviated by more than two orders of magnitude, reaching no consensus in the dominant external cycling route or atmospheric variability in the external cycling[4,21,32,34,38].

The $NO_2$ mixing ratios ranged from 10 pptv to 14.2 ppbv, with median values of approximately 30 pptv in the FT and 218 pptv in the BL (Fig. 2 and Table S2). While ppbv levels of $NO_2$ were occasionally found as we flew through urban and industrial plumes, the median values of both HONO and $NO_2$ were not affected by these high values and were still representative of the background troposphere. The median $NO_2$ in the BL and the FT agreed with the established $NO_2$ distribution in these background atmospheres[5,21,26,38,40–42]. The median HONO mixing ratio was approximately 7.0 pptv in the FT background and 12.1 pptv in the BL background (Fig. 2 and Table S2). Our data provided the first illustrations of the HONO distribution in such a variety of low-$NO_x$ atmospheres. The median HONO in the BL was among the lowest ever reported in specific low-$NO_x$ environments, such as in snow/ice-covered polar areas[20,27–29], pristine terrestrial boundary layers[32,43], and clean marine boundary layers[4,23,25,38]. The median HONO in the FT was slightly lower than values reported from other aircraft observations in low-$NO_x$ atmospheres[41,43] but was one or two orders of magnitude lower than those observed in plumes[34,44].

The GEOS-Chem model was used to provide a benchmark for the concentration ratios among reactive nitrogen species, such as the HONO/$NO_y$ ($\equiv NO_x + NO_z$) ratio, $NO_2$/$NO_y$ ratio and HONO/$NO_2$ ratio. The emission inventory applied in GEOS-Chem overestimated $NO_x$ emissions[45]. However, this drawback might not interfere with the model simulation of the HONO/$NO_2$ ratio and $NO_2$/$NO_y$ ratio in various low-$NO_x$ air masses since the reactive nitrogen species aged to reach their photosteady state (PSS) in the low-$NO_x$ atmosphere. Even with the anticipated model overestimation of $NO_2$ and consequently overestimation of $NO_z$ (Fig. S2), the HONO concentration, HONO/$NO_y$ ratio and $NO_2$/$NO_y$ ratio were substantially underestimated in the model (Fig. 2), indicating the external cycling of $NO_x$. Moreover, model underestimation of the HONO/$NO_2$ ratio and HONO/$NO_y$ further revealed that HONO might be an intermediate in the external cycling of $NO_x$, as HONO regenerated in the external cycling would rapidly photolyze to produce $NO_x$ (Fig. 2). It was speculated that the relatively slow turnover rate of $NO_x$ to produce HONO was associated with a low HONO/$NO_2$ ratio (<0.05, as commonly observed in high-$NO_x$ atmospheres[9]) to balance the rapid turnover rate of HONO to produce $NO_x$ via HONO photolysis and the Leighton cycle. Herein, a much higher HONO/$NO_2$ ratio relative to the PSS prediction in GEOS-Chem indicated a net turnover of HONO to produce $NO_x$ in the external cycling and presented HONO as an intermediate product. Nevertheless, direct regeneration of $NO_x$ in the external cycling bypassing HONO intermediate could not be excluded.

The external cycling route of $NO_x$ with $NO_z$ as a precursor and HONO as an intermediate could be proposed based on the synthesized evidence from the concentration ratios of reactive nitrogen species. Anthropogenic emission perturbation, internal cycling mechanisms, and measurement interferences to reconcile the model–observation discrepancies could be safely excluded. First, anthropogenic emissions of HONO and $NO_x$ perturbed the distribution of HONO within a transport height of approximately 300 m while perturbing the distribution of $NO_2$ within the boundary layer[16,19,32]. The nineteen research flights spent over 85% of the time 600 m above the ground surface in the BL and over 45% of the time in the FT. Hence, anthropogenic

emissions of HONO and $NO_x$ could be excluded as the major reason for the model–observation discrepancies in the concentration ratio of reactive nitrogen species. Second, the photosensitization reactions of $NO_2$ produced HONO, with photo-enhanced rates several-fold higher at noon in low-$NO_x$ conditions than at nighttime in high-$NO_x$ conditions;[7,8,10] however, they were still too slow to account for the high HONO/$NO_2$ ratio observed (Fig. 2). In fact, fully reconciling model–observation discrepancies in the HONO/$NO_2$ ratio required a reactive uptake coefficient of $NO_2$ to produce HONO to be in the order of magnitude of $10^{-3}$, which was nearly two orders of magnitude higher than the proposed photo-enhanced values at noon in low-$NO_x$ conditions[7,10]. In addition, such photosensitization reactions were not related to the model–observation discrepancy in the $NO_2$/$NO_y$ ratio. Finally, a substantially positive measurement interference in HONO coupled with negative measurement interferences in $NO_z$ species, such as $HNO_3$ and $pNO_3$, might improve the model–observation agreement for both the HONO/$NO_2$ ratio and $NO_2$/$NO_y$ ratio. However, such species-dependent interference was not practical, as any positive (or negative) nitrite anion interference in HONO measurement would also result in a positive (or negative) interference in $pNO_3$ and $HNO_3$ measurements. A comparison of our LPAP HONO measurements with those by the DOAS instrument showed reasonable consistency in the concentration range beyond the detection limit of both instruments, but DOAS showed a smaller value relative to LPAP in the lower concentration range[4]. Potential chemical interferences for LPAP HONO, including $HNO_4$ and particulate nitrite, were not directly measured. PSS $HNO_4$ interference calculations suggested that it only caused minor interference (<15% of signal)[32]. The low partitioning ratio of particulate nitrite over HONO and low sampling efficiency of particulate nitrite in the LPAP system also suggested minor interferences (<1% of signal)[46–48]. Therefore, our HONO measurement had only been corrected for PSS $HNO_4$ interference and therefore, the LPAP HONO measurement appeared to be reasonably reliable in our campaign. Measurements of $pNO_3$ and $HNO_3$ had not been corrected for PSS $HNO_4$ interference since the interference was small relative to the signals. Similarly, reliable $NO_x$ measurements have been widely reported in the literature[40]. Although potential positive interference was not totally excluded[49], model underestimation of such as the HONO/$NO_2$ ratio would be even worse assuming potential positive interference of $NO_2$. As such, although measurement interferences could not be completely excluded for HONO, $NO_x$, $pNO_3$, and $HNO_3$ measurements, the potential interferences were either very small or did not show the species-dependent characteristics required to reconcile the model–observation discrepancies. These analyses further supported the external cycling as a proper cause to reconcile the model–observation discrepancies.

Helas and Warneck and Ye et al. deduced that the lack of expected daytime minima of $NO_2$ and HONO might be additional critical evidence for the external cycling, at least for the low-$NO_x$ marine boundary layer[4,26]. Bell-shaped diurnals of HONO and $NO_2$ were observed as critical evidence for the external cycling powered by snow photochemistry in polar areas[29,50]. Their deduction followed a budget analysis of $NO_2$ and HONO. The oxidation of $NO_2$ was a major sink for $NO_2$, with the rate scaling with the radical concentration or photolysis frequency of $O_3$ ($jO^1D$). Additionally, partitioning of $NO_2$ over NO was also promoted by fast photolysis of $NO_2$ to produce NO at noon, as determined in the Leighton cycle[40,42]. Both chemical processes contributed to a noontime minimum of $NO_2$. For HONO, daytime photolysis was the largest budget term and dominated the diurnal profile[1,2]. Therefore, typical U-shaped diurnal profiles for $NO_2$ and HONO were expected and generally observed in high-$NO_x$ atmospheres[1,2,40,42]. External cycling of $NO_2$ and HONO might compensate for their daytime losses and result in atypical diurnal profiles of HONO and $NO_2$, including flat and even bell-shaped profiles depending on the rate of external cycling[4,22,25].

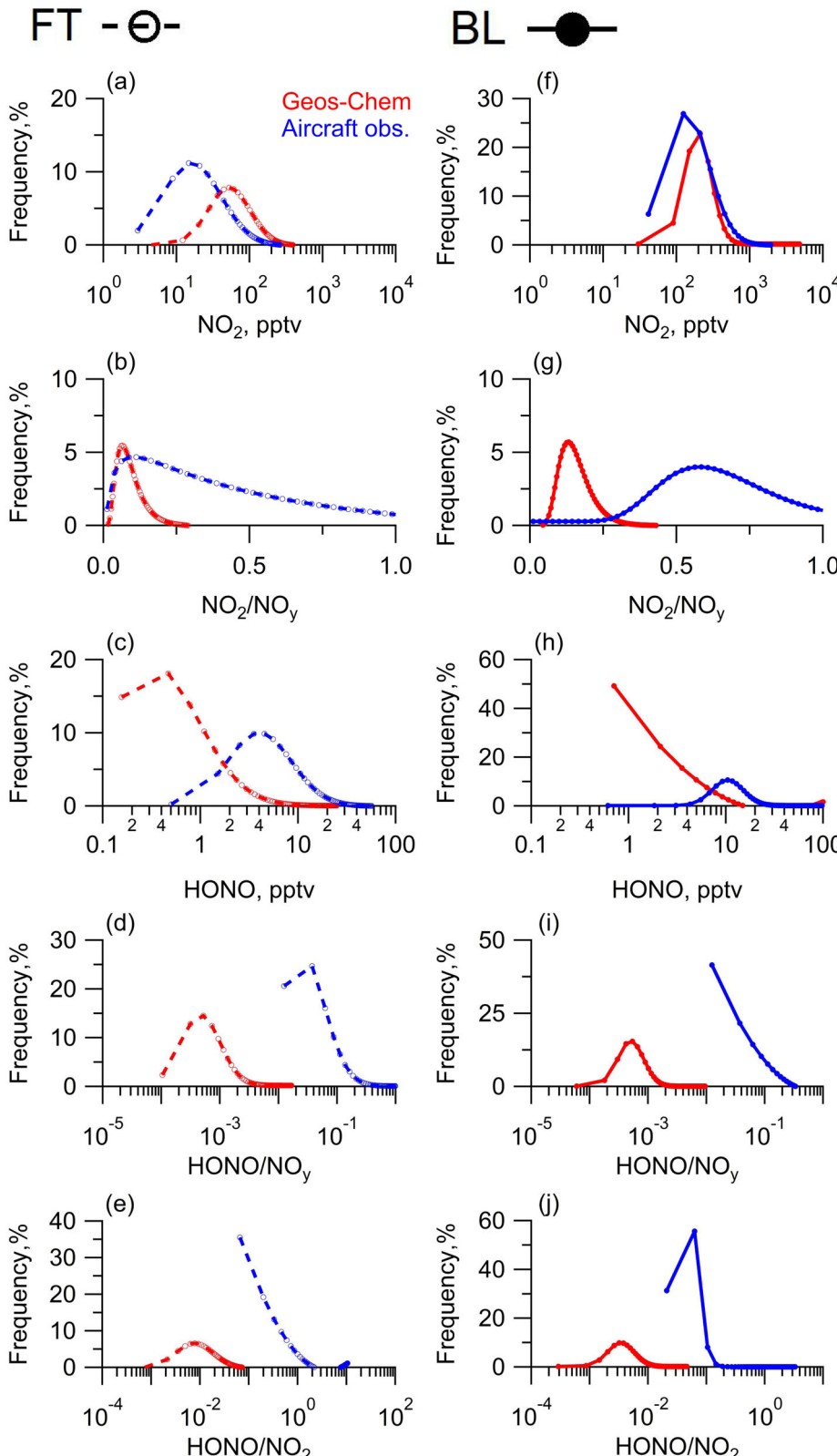

**Fig. 2 | Frequency distribution of concentrations and concentration ratios of the reactive nitrogen species as measured onboard the C-130 research aircraft and simulated by GEOS-Chem. a–e** Distribution of $NO_2$ concentration, the concentration ratio of $NO_2/NO_y$, HONO concentration, the concentration ratio of $HONO/NO_y$, and the concentration ratio of $HONO/NO_2$ in the free troposphere (FT), respectively. **f–j** Distribution of $NO_2$ concentration, the concentration ratio of $NO_2/$ $NO_y$, HONO concentration, the concentration ratio of $HONO/NO_y$, and the concentration ratio of $HONO/NO_2$ in the boundary layer (BL), respectively. The red lines with circles represent the GEOS-Chem model predictions. The blue lines with circles represent our aircraft observations. Open circles and solid circles represent data points from the FT and the BL, respectively. Dashed lines and solid lines connect adjacent data points.

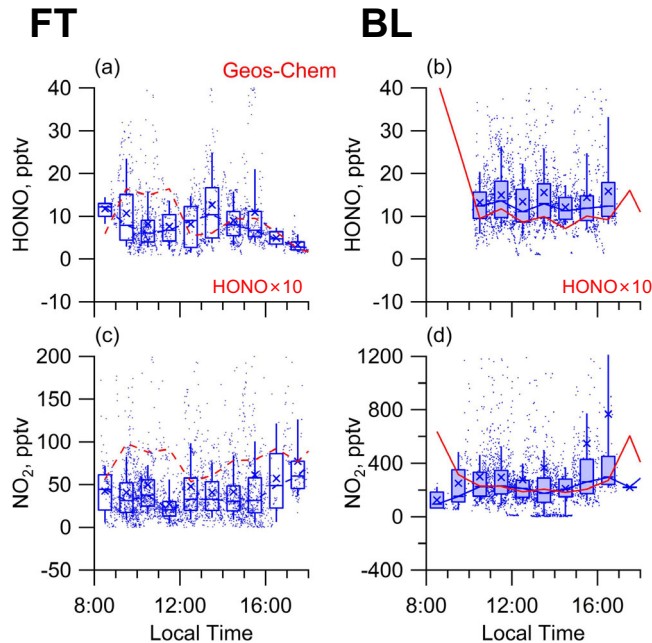

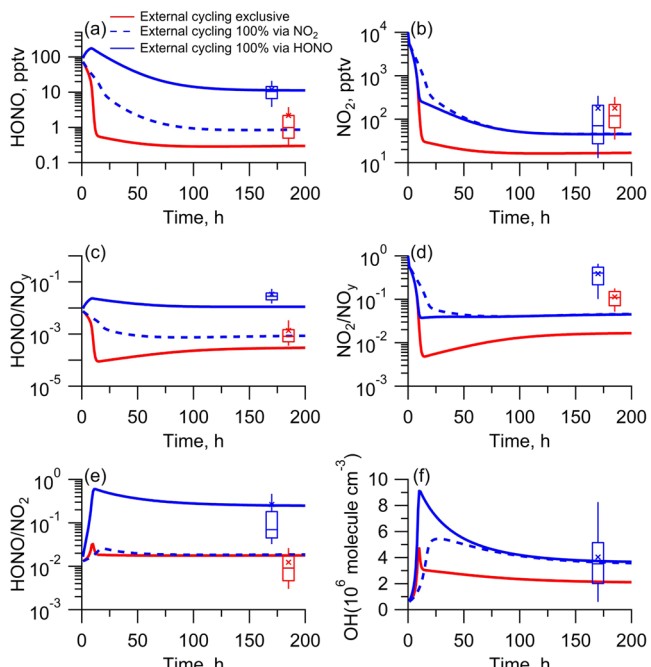

**Fig. 3 | Diurnal profiles of HONO and NO₂ as measured onboard the C-130 research aircraft and simulated by GEOS-Chem. a, b** Diurnal profiles of HONO in the free troposphere (FT) and boundary layer (BL), respectively. **c, d** Diurnal profile of NO₂ in the FT and BL, respectively. The red dashed and solid lines represent the median values of the GEOS-Chem model predictions. The blue boxes represent, from top to bottom, the 75th, 50th, and 25th percentiles; the whiskers above and below the boxes represent the 90th and 10th percentiles; and the cross represents the mean value of our aircraft observations. Open boxes and solid boxes represent data points from the FT and the BL, respectively. The blue dashed lines and solid lines connect adjacent median values.

**Fig. 4 | Distribution patterns of concentration and concentration ratios of reactive nitrogen species, and OH radicals during plume aging as simulated by the MCM model. a, b** Concentrations of HONO and NO₂, respectively. **c–e** Concentration ratio of HONO/NO$_y$, NO₂/NO$_y$, and HONO/NO₂, respectively. **f** Concentration of OH radicals. The red line represents the model S0, which excludes external cycling. The blue line and blue dashed line present the model S1 and S2, which include the proxy mechanism for external cycling with HONO yields of 100% (0% yield for NO₂) and 0% (100% yield for NO₂), respectively. The blue and red boxes represent the results of our aircraft observations and GEOS-Chem simulations, respectively. From top to bottom of the box, the 75th, 50th, and 25th percentiles are shown; the whiskers above and below the boxes represent the 90th and 10th percentiles; and the cross represents the mean values.

Although the 19 research flights sampled various airmasses over a large geographic area across a variety of chemical regimes in the atmosphere during different hours of the day, typical photochemical peaks in radicals and jO¹D were still observed (Fig. S3), and therefore the typical diurnals of HONO and NO₂ were expected. However, we observed atypical diurnal profiles that lacked daytime minima of NO₂ and HONO in more general BL, adding to the previous observation of the same kind in the clean marine boundary layer and polar boundary layer and first in the FT (Fig. 3). This observation also differed from the GEOS-Chem simulation which excluded external cycling and simulated the expected diurnal profiles of NO₂ and HONO, especially in the BL (Fig. 3). Therefore, the atypical diurnal profiles of NO₂ and HONO provided further observational evidence of the temporal distribution of reactive nitrogen to verify the external cycling of NO$_x$ and HONO. Notably, neither of the internal cycling mechanisms, such as the photosensitization reaction of NO₂, nor potential measurement interferences of HONO or NO₂, were able to reconcile model–observation discrepancies in the diurnal profiles of NO₂ and HONO.

**Chemical model evaluation of external cycling**
As the GEOS-Chem did not compile detailed radical chemistry along the oxidation mechanism of VOCs[51,52], a nearly explicit chemical model (MCM version 3.3.1, http://mcm.leeds.ac.uk/MCM/) was adapted to simulate the photochemical evolution of a random power plant plume (Fort Martin station power plant plume captured in RF10, Methods) to represent nitrogen cycling photochemistry across various NO$_x$ regimes over a large geographic area. The objectives of the chemical model simulation were to explore the fundamental characteristics of external cycling (i.e., HONO being an intermediate, determinative role of external cycling in the observed high ratio of HONO/NO₂ and NO₂/NO$_y$, and the perturbation of external cycling on OH photochemistry)

along with the aging of the plume. To be more specific, measurements in the power plant plume were chosen to initialize our model, while chemical conditions in the low-NO$_x$ atmosphere were carefully summarized to constrain the model. To focus on the chemical evolution of composition in the plume, a conceptual photochemical evolution under solar noon conditions was simulated in our model. To avoid discussion on any specific external cycling mechanism and related arguments on the reaction rate of specific external cycling routes, a proxy mechanism for the external cycling employing pNO₃ as a representative of NO$_z$ species and the precursor of HONO and NO$_x$ in the external cycling was included in the chemical model scheme. The pNO₃ photolysis rate or the reaction rate of a general NO$_z$ species was set up based on the assumption that the external cycling could fully account for the unknown source of HONO or NO₂. Previous field observations or laboratory measurements of the photolysis rate constant were not referred to or compared with, as only a proxy mechanism, rather than a specific mechanism, of the external cycling was the very core of the discussion. Three independent models were run: one excluded the external cycling of NO$_x$ and HONO (model S0), and the other two included external cycling, with/without HONO as a NO$_x$ intermediate (model S1-S2). Since the external cycling was verified and HONO was identified as an intermediate product based on our observations, chemical model S1 was expected to best represent our observations on the distribution of HONO and NO₂ in varied NO$_x$ regimes.

The kinetic curve of reactive nitrogen species during the aging of the Fort Martin station power plant plume was shown in Fig. 4. NO₂ went through quasi-exponential decay in the initial period and then

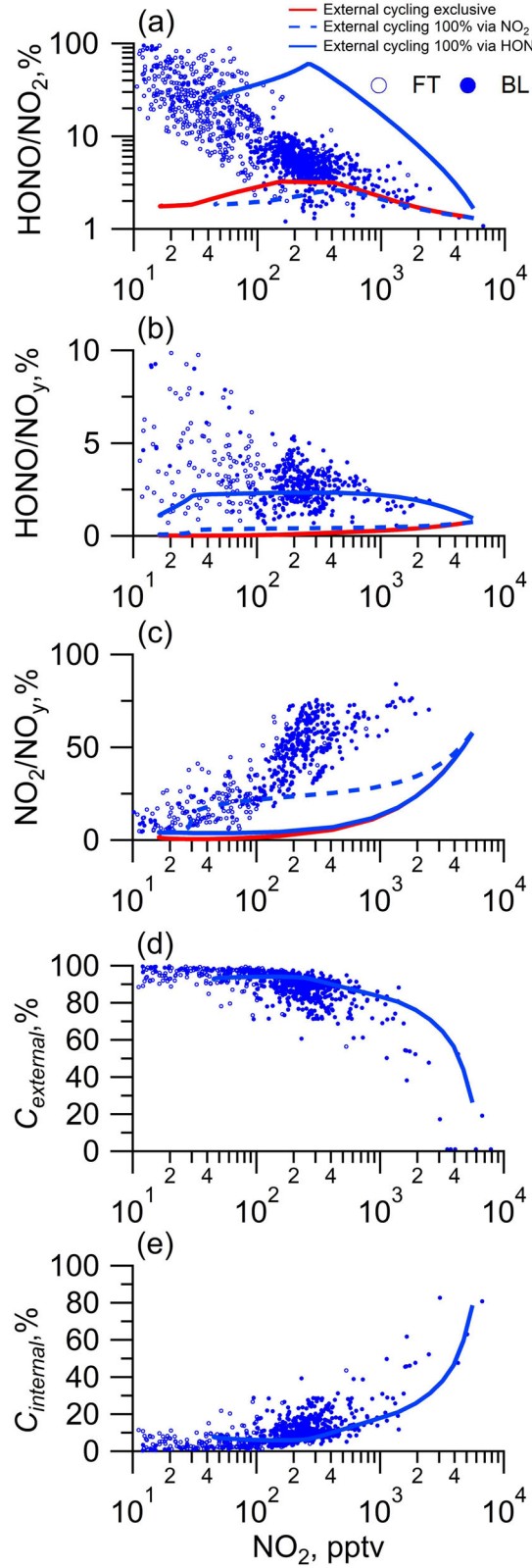

**Fig. 5 | Scatter plots of the concentration ratios of reactive nitrogen species, and contribution of external and internal cycling, against NO₂, as measured onboard the C-130 research aircraft and simulated by the MCM model.**
**a–c** Concentration ratios of HONO/NO₂, HONO/NO_y, NO₂/NO_y, respectively.
**d**, **e** External source contribution ($C_{external}$) and internal source contribution ($C_{internal}$), respectively. Open circles and solid circles represent measured data points from the free troposphere (FT) and the boundary layer (BL), respectively. The red line represents model S0, which excludes external cycling. The blue line and blue dashed line present the model S1 and S2, which include the proxy mechanism for external cycling with HONO yields of 100% (0% yield for NO₂) and 0% (100% yield for NO₂), respectively.

regeneration of $NO_2$ from the external cycling approached an equilibrium state, resulting in relatively consistent $NO_2$ levels that had been persistently observed in various low-$NO_x$ atmospheres in our study and in the literature[4,21,40].

The counterbalancing role of $NO_x$ regeneration in the external cycling extended the $NO_x$ lifetime and sustained comparable $NO_x$ abundance in aged airmasses or remote atmospheres. It also challenged the current chemical model scheme involving the continuous oxidative decay of $NO_x$ and a small $NO_x$ regeneration rate, leading to extremely low $NO_x$ abundance in the tropospheric background. In model S0, the $NO_2/NO_y$ ratio was not negligible but was extremely low (Fig. 4) due to the low regeneration rate of $NO_x$ via transport and the thermal decomposition of PAN, even with higher PAN and higher rates of PAN decomposition applied in the model than those obtained from our aircraft observations (Fig. S4). The external cycling of $NO_x$ in model S1 or S2 better agreed with the observed $NO_2$ concentrations and concentration ratios of $NO_2/NO_y$ in low-$NO_x$ environments, confirming the determinative role of the external cycling in the distribution of $NO_2/NO_y$ ratio especially in low-$NO_x$ atmospheres. Model S1 also better agreed with the HONO concentration, HONO/$NO_y$ ratio, and HONO/$NO_2$ ratio, confirming HONO as an intermediate in the external cycling. Compared to other $NO_x$ regeneration mechanisms, such as thermal decomposition and photolysis of PAN (gaseous $HNO_4$, organic nitrates, and nitric acid), the external cycling involving HONO as an intermediate proceeded far more rapidly, especially in low-$NO_x$ atmospheres.

To note, our observation and modeling consistently demonstrated the environmental variability pattern in the role of external cycling as the plume aged, and our modeling with the external cycling better captured the observed ratios of HONO/$NO_2$, HONO/$NO_y$, $NO_2/NO_y$ as $NO_x$ decreased (Fig. 5). These environmental variability patterns were mainly rationalized by the accumulation of $NO_z$ (the external cycling precursor) as the plume aged. Consequently, an increasing trend in the HONO/$NO_2$, HONO/$NO_y$ ratio and therefore a relatively strong external cycling contribution ($C_{external}$) to the budget of HONO and $NO_x$ as $NO_x$ decreased were observed (see Calculation of $C_{internal}$ and $C_{external}$ in Method section). The environmental variability of the rate constant of the external cycling, i.e., the high rate constant of the external cycling in low $NO_x$ atmospheres, might also contribute to the environmental variability patterns[30]. Although the environmental variability in the rate constant of the external cycling was not considered in our box model scheme, our model reasonably captured the observations across $NO_x$ regimes. Therefore, both our observations and the model simulation illustrated increasing perturbations on atmospheric budgets of reactive nitrogen species by the external cycling as $NO_x$ decreased.

Frankly speaking, it is difficult to comprehend the observed trend of $C_{external}$ as $NO_x$ decreases since it goes against the traditional theories in which internal cycling dominates the HONO budget. The shifting of the HONO/$NO_2$ ratio as $NO_x$ decreased in the high-$NO_x$ atmosphere was so slow and limited in a narrow range, typically from 0.01 to 0.1 (Figs. 4, 5), that it might have been considered unchanged given the observational uncertainties. The relationship between HONO and $NO_x$ was strengthened by numerous observations in high-$NO_x$

approached a steady period with a stabilized $NO_2/NO_y$ ratio and any other concentration ratios of reactive nitrogen species (Fig. 4). The initial period simulated reactive nitrogen photochemistry in a fresh plume or high-$NO_x$ atmosphere where the oxidation of $NO_x$ to form $NO_z$ species dominated the $NO_x$ budget, leading to rapid $NO_2$ decay. The steady period mimicked reactive nitrogen photochemistry in aged plumes or in low-$NO_x$ atmospheres where the oxidation loss and

atmospheres[1,9,13], and the dominant role of the internal cycling tended to be mistakenly extrapolated to low-NO$_x$ chemical regimes as a consequence. Our characterization of the chemical-regime-dependent contribution of the external cycling better summarized the environmental variability feature of the external cycling and properly addressed the observational argument regarding missing HONO sources in high-NO$_x$ atmospheres and low-NO$_x$ atmospheres[44].

The increasing contribution of the external cycling to the budget of HONO and NO$_x$ as NO$_x$ decreased might have amplified its perturbation on OH photochemistry because OH production was mostly sensitive to the abundance shifting of NO$_x$ in the low-NO$_x$ atmosphere. In addition, the daytime compensation of NO$_x$ by the external cycling might have further amplified its perturbation role because the photochemical production of OH overlapped with the external cycling of NO$_x$ and HONO during the daytime. This point was illustrated in the model underestimation of OH. Due to model S0 omission of the external cycling, the model underestimated OH, and this underestimation was more apparent in a low-NO$_x$ atmosphere than in a high-NO$_x$ atmosphere (Fig. 4). The model underestimation of OH was also confirmed to be more closely linked with primary OH production via photolysis of HONO (driven mainly by the internal cycling) in a high-NO$_x$ atmosphere but more closely associated with NO$_x$-catalyzed secondary OH production driven by the external cycling of NO$_x$ in a low-NO$_x$ atmosphere. The S0 model underestimation of OH abundance reached approximately 41% in a low-NO$_x$ atmosphere (Fig. 4), comparable to a previous CTM evaluation, which showed a perturbation of the NO$_x$ budget and therefore the OH budget by approximately 40% via a proxy cycling mechanism in the marine boundary layer[35].

Overall, the results of our analysis indicate the dominant role of external cycling in the chemical budget of reactive nitrogen in low-NO$_x$ atmospheres and its significant impact on oxidant photochemistry, thus calling for an urgent and significant revision in our understanding of the atmospheric chemistry of reactive nitrogen and oxidants. To advance this goal, extensive research efforts are needed, for example, laboratory work to better parameterize the kinetics and mechanisms of dominant external cycling routes, including but not limited to pNO$_3$ photolysis, and field measurements to establish spatial and temporal distributions of reactive nitrogen species that are only sporadically available at this time, especially for low-NO$_x$ atmospheres[5,17,34,53,54].

## Methods
### Aircraft observation
A total of 19 research flights were conducted, mostly from 8:00 to 18:00 local time, in various low-NO$_x$ troposphere environments. The raw data from the first 15 min after taking off and the last 15 min before landing were excluded from the analysis to avoid interference from pollution at the airport and were then averaged for 3 min for further analysis.

O$_3$ and NO$_x$ were measured using chemiluminescence methods[55,56]. HONO, HNO$_3$, and pNO$_3$ were measured using a wet-chemistry method similar to LOPAP[57]. Aerosol number-size distributions were measured using a scanning mobility particle sizer (SMPS) and an ultrahigh sensitivity aerosol spectrometer (UHSAS)[58]. Alkyl nitrates, PANs, and VOCs were measured by a trace organic gas analyzer (TOGA)[59]. Free radicals were measured using a selected-ion chemical-ionization mass spectrometer (SICIMS)[60]. Photolysis frequencies were calculated from measurements of a scanning actinic flux spectroradiometer (SAFS)[61].

### Calculation of C$_{internal}$ and C$_{external}$
The internal cycling included heterogeneous reactions of NO$_2$ on ambient surfaces, gas-phase reaction between NO and OH radicals, HONO photolysis, and gas-phase reaction between HONO and OH

radicals. The chemical reactions are listed as follows.

$$NO_2 + surface \rightarrow HONO + other\ products$$

$$HONO + h\upsilon\,(\lambda < 400\,nm) \leftrightarrow OH + NO$$

$$HONO + OH \rightarrow H_2O + NO_2$$

The internal source of HONO ($P_{internal}$) is calculated with the equation below:

$$P_{internal} = k_{R1}[NO_2] + k_{R2}[OH][NO] \tag{1}$$

where [X] represents the concentration of species X in molecule cm$^{-3}$; $k_{R2}$ is the rate constant of NO with OH radicals; $k_{R1}$ is defined as the heterogeneous uptake rate constant of NO$_2$ on the aerosol surface and is calculated with Eq. (2).

$$k_{R2} = \frac{1}{4} \times s/\upsilon \times c \times \gamma \tag{2}$$

where $s/\upsilon$ represents the aerosol surface density, which is calculated from the particle number-size distribution by the SMPS. $c$ represents the average molecular speed of NO$_2$. $\gamma$ represents the uptake coefficient of NO$_2$ on an aerosol particle surface. An upper limit of 10$^{-4}$ was adopted to estimate the upper limit of internal sources of HONO[62].

With a photolytic lifetime of approximately 10 min, the photosteady-state assumption is applicable for HONO. The external source of HONO ($P_{external}$) is calculated according to the photosteady-state budget as follows.

$$P_{internal} + P_{external} = (j_{HONO} + k_{R3}[OH]) \times [HONO] \tag{3}$$

where [X] represents the concentration of species X in molecule cm$^{-3}$, $j_{HONO}$ represents the photolysis frequency of HONO, and $k_{R3}$ represents the rate constant of HONO with OH radicals. Soil and anthropogenic emissions of HONO are not included in the budget. The aircraft measurements were mostly taken $\geq$ 600 m above the ground in the boundary layer and up to 8000 m in the free troposphere. The air masses encountered were therefore decoupled from ground surface HONO sources such as soil emissions and heterogeneous reactions of NO$_2$ at ground surfaces[63]. The high HONO/NO$_2$ ratio observed here also excluded influences from primary HONO emissions except for occasional influences from power plant plumes or city plumes. Therefore, only in situ chemical reactions were considered for the internal and external cycling rate calculation in this study.

The relative contribution of the internal source and external source to the total HONO source, $C_{internal}$ and $C_{external}$, can be defined as follows:

$$C_{internal} = \frac{P_{internal}}{P_{internal} + P_{external}} \tag{4}$$

$$C_{external} = \frac{P_{external}}{P_{internal} + P_{external}} \tag{5}$$

### Nearly explicit chemical model simulation
The master chemical mechanism (MCM v3.3.1) is adapted here to simulate nitrogen cycling photochemistry and its impacts on the oxidative capacity of the atmosphere. The specific objective of the model run was to test whether the external cycling proxy mechanism, i.e.,

particulate nitrate photolysis or reaction of a general $NO_z$ species, can reproduce the unique distribution patterns of OH, HONO, and $NO_2$ in the low-$NO_x$ troposphere.

The chemical scheme extracted from the MCM website was employed. The initial mechanism revisions of the MCM model include the addition of the heterogeneous production of HONO from $NO_2$ reactions, dry deposition-driven removal of model-calculated OVOC intermediates and reactive nitrogen species, and a transport source for $HNO_3$ and PAN to compensate for dry deposition losses of $NO_y$ species[22,25]. The conversion rate of HONO from $NO_2$ reactions is set to be $1.4 \times 10^{-5} s^{-1}$ (~0.05 h$^{-1}$). The dry deposition-driven removal of model-calculated OVOC intermediates and reactive nitrogen species are set to be $9.0 \times 10^{-6} s^{-1}$ and $1.0 \times 10^{-5} s^{-1}$, respectively. Both the transport source rate for $HNO_3$ and PAN are 18 pptv h$^{-1}$. This model mechanism setup is referred to as the baseline model, S0, which excludes the external cycling proxy mechanism. In further revised models (model S1 and S2), we added HONO production from the proxy mechanism of the external cycling (Table S3). The production rate of HONO or $NO_2$ ($P_{pNO_3}$) was expressed as follows:

$$P_{pNO_3} = [pNO_3] \times j_{HNO3} \times EF \qquad (6)$$

where $j_{HNO3}$ is the photolysis frequency of $HNO_3$ in its gaseous form, and the enhancement factor, $EF$, is the ratio of the photolysis rate constant of particulate nitrate relative to $j_{HNO3}$. A median $EF$ of 150 and a pNO$_3$/tHNO$_3$ partitioning ratio of 0.5 were employed in the model to represent the average conditions for the background troposphere[30]. A HONO yield of 100% (S1, 0% yield for $NO_2$) or 0% (S2, 100% yield for $NO_2$) was adapted in models S1 and S2, respectively.

proxy mechanism in Model S1 : $\quad pNO_3 + h\nu \xrightarrow{150 \times j_{HNO3}} HONO$

$$\xrightarrow{j_{HONO}} NO_2$$

proxy mechanism in Model S2 : $\quad pNO_3 + h\nu \xrightarrow{150 \times j_{HNO3}} NO_2$

To mimic the photochemical evolution of reactive nitrogen species in the typical tropospheric background and represent external cycling and its perturbation on OH photochemistry across various $NO_x$ regimes over a large geographic area, the model was initialized with a $NO_x$ mixing ratio of 10 ppbv, as observed in the Fort Martin station power plant plume. Concentrations of $O_3$ and OH were initialized at 30 ppbv and $1.7 \times 10^6$ cm$^{-3}$, respectively, which were typically observed in our daytime measurements. To best represent the radical chemistry for the background troposphere, an OH reactivity of approximately 5 s$^{-1}$ was used for the model based on our aircraft measurements of VOCs and other species that consume OH, such as $O_3$, $NO_x$, and CO. CO, $CH_4$, HCHO, and $CH_3CHO$ were found to be the top four contributors to OH reactivity in the background troposphere. To simplify the model, all measured VOCs, except HCHO and $CH_3CHO$, were grouped with CO to obtain an equivalent OH reactivity. The oxidative mechanism of these grouped VOCs was thus not included in our model. The photosteady state was established quickly in the model under these initial conditions. Because our focus was on the photochemical cycling of HONO and $NO_2$, the models were run without day-night shifts and had constant photolysis frequencies under zero solar zenith angle conditions (Table S4) to mimic the continuous 400 hr photochemical aging of the power plant plume in the background troposphere. After approximately 100 hours, the model simulation reached a stable stage. Thus, only results for the first 200 hours are shown. The model settings and initialization conditions are summarized in Tables S3 and S4.

Our model generally reproduces the observation of reactive nitrogen in various atmospheric chemical regimes. The slightly lower modeling of the HONO/$NO_y$ ratio and $NO_2$/$NO_y$ ratio might be partly accounted for by measurement underestimations in $NO_y$ species

(Figs. 5 and S4). Air bubbles can deactivate the Cd columns in LPAPs, leading to potentially lower nitrate-to-nitrite conversion efficiencies; measurement underestimations in pNO$_3$ and $HNO_3$ from this phenomenon were recently estimated to reach twofold[23]. Incomplete $NO_y$ measurements in the aircraft platform, such as measurements of other alkyl nitrates, and uncertainties in the organic nitrogen formation mechanism might be other reasons[17]. An assumed extreme case of 100% HONO yield in model S1 might be another reason, as the direct yield of $NO_2$ cannot be excluded[4,25].

## GEOS-Chem simulation

The GEOS-Chem model (version 9-02; www.geos-chem.org) regional simulations were conducted in a one-way nested grid formulation with the native GEOS-5 Forward Processing horizontal resolution of 0.25° × 0.3125° over the North America domain (130–60° W, 10–60° N). The initial and boundary conditions were obtained through a global GEOS-Chem model simulation with a coarser resolution of 2° × 2.5° (reduced from the native GEOS-5 forward processing grid). The GEOS-Chem meteorological fields were obtained from the assimilated products of the NASA Global Modeling and Assimilation Office Goddard Earth Observing System (http://gmao.gsfc.nasa.gov/products/). The model simulations had 47 vertical layers in the atmosphere. We used a non-local planetary boundary layer mixing scheme developed by Holtslag and Boville (1993) and implemented in GEOS-Chem by Lin and McElroy (2010)[64,65]. The tropospheric chemistry mechanism was described in Parrella et al. (2012)[66]. No further updates, including external cycling, were made. Emissions of atmospheric components included anthropogenic emissions from the Emissions Database for Global Atmospheric Research (https://edgar.jrc.ec.europa.eu), biogenic emissions of volatile organic carbons from the Model of Emissions of Gases and Aerosols from Nature inventory[67], fire emissions from the Global Fire Emissions Database[68], soil $NO_x$ emissions[69], and lightning $NO_x$ sources[70]. The dry deposition scheme used a resistance in-series model based on Wesely (1989)[71]. The wet deposition scheme included the scavenging of soluble tracers in convective updrafts as well as the rainout and washout of soluble tracers[72]. Simulation results corresponding to the research flight track during the same period were extracted for measurement comparison.

## Data availability
The field data used in this study have been deposited and are freely available in the SAS project data archive (http://data.eol.ucar.edu/master_list/?project=SAS).

## Code availability
The full set of the MCM model mechanism can be extracted from the MCM website (https://mcm.york.ac.uk/MCM/) and the subset is also available from the corresponding author upon request.

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

## Acknowledgements

This analysis and model work is supported by the National Natural Science Foundation of China (grants 41875151, C.Y.). We acknowledge the U.S. National Natural Science Foundation (grants AGS−1216166 and AGS−1826989, X.Z.) and members of the SAS project for supporting and providing the field measurements. We really appreciate Xueling Meng for her kind help in preparing Fig. 1.

## Author contributions

C.Y. interpreted the data and wrote the manuscript with revision advice from X.Z., J.W., C.Z., R.W.M., C.C., R.L.M., T.C., R.S.H., J.O., E.C.A., J.H., S.H., K.U., A.W., J.S., T.K., J.N.S. and A.G., Y.Z. prepared the figures. Y.W. performed MCM model calculation. S.S. performed the GEOS-Chem simulation.

## Competing interests

The authors declare no competing interests.
