## [Peer Review File · Nature Communications]

Synthesizing evidence for the external cycling of NO_x in high-
to low-NO_x atmospheresReviewer #1 (Remarks to the Author):

Ye et al. use aircraft measurements of nitrogen oxides (NO_x), nitrous acid (HONO) and the sum of oxidized reactive nitrogen species (NO_y=NO_x + reservoirs) to show that recycling of NO_x from the reservoir species is faster than what is commonly accounted for in atmospheric chemistry models and that it involves HONO as an intermediate species. The authors show that including the fast recycling of NO_x in a model improves leads to a substantial increase in NO₂ and HONO concentrations in free troposphere in the model, making it consistent with aircraft observations. This also leads to a substantial increase in the modeled OH concentration.

Previous studies have shown that recycling of NO_x from its reservoirs is faster than what is predicted by established theory, but the mechanism and rate of the recycling reactions are not clear. In this study, the authors analyze the complete set of observations from a regional aircraft campaign to provide further evidence of NO_x recycling. Previously, they had published similar results based on a smaller subset of the observations. The study uses high-quality observations, well-established models, and appropriate analysis to strengthen the evidence for NO_x recycling.

The manuscript lacks clarity, key sections are missing, and is difficult to follow. Additionally, the results, as presented, do not constitute a significant advance on this topic. However, the study can be improved by considering the following revisions:

1. The manuscript is missing a discussion of the results of this work in comparison to previous studies on this topic. Specifically, how does the recycling rate compare to what has been reported in the literature and can it be explained by mechanisms that have been proposed previously? The authors themselves have published several studies on particulate nitrate photolysis as the potential mechanism, but for some reason seem reluctant to discuss this in the main text, although it is discussed in the supplement. The significance of this study depends on whether it gives an insight on these questions.
2. The modeling of the power plant plume (lines 226 onwards and fig. 4) is confusing. It is not clear how a "plume" would even be defined after 150-200 hours of transport and dilution with the background air. To avoid confusion, it would be clearer if only the results from the box model at steady-state were presented and compared to aircraft observations in Fig. 4.
3. The analysis related to the effect of NO_x recycling on OH concentrations (Lines 306-319 and Fig. 4) is inadequate. It is based on results of modeling an emission plume that has travelled 150-200 hours, which presents certain difficulties (as mentioned in point 2). In addition, it appears that not all of the OH production and loss terms were taken into account. For instance, the OH loss rate was set to a fixed value based on the measurements of the primary OH reactants, but this probably underestimates the OH reactivity (see Thames et al., 2020, 10.5194/acp-20-4013-2020).
4. The section from lines 168 to 192 discusses the possibility of measurement artifacts for HONO, HNO₃ and particulate nitrate. Additionally, there is a known positive interference in the NO₂ chemiluminescence measurements used in this study. I recommend that this issue also be addressed within this section.
5. The manuscript is lacking a crucial section - the "Methods" - which is only available in the supplement. This makes it challenging to interpret the results. Please add the "Methods" section to the main text. Specifically, the authors should provide a more detailed explanation of the box modeling approach and the different model configurations. The calculation of the C(external) source in configurations S1 and S2 is not clearly explained and needs to be addressed.
6. The introduction could benefit from a more detailed and specific description of the problem addressed in the manuscript, including an explanation of the relevant chemistry. Fig. 1 may not be necessary and could potentially be removed.
7. Minor/technical comments:
 - a. The title of the manuscript does not accurately reflect the main findings of the study, which focus on NO_x recycling rather than the oxidative capacity of the atmosphere.

- b. Line 54: hydroxyl radical
- c. Line 74-75: NO_x can also be recycled from alkyl nitrates, HNO₄, CH₃ONO₂, and HNO₃.
- d. Line 86: It is not obvious that a regional campaign in one season can provide a "global view". Please clarify or consider rewording.
- e. Lines 91-99: Measurement details need to be included in the main text in the Methods section.
- f. Line 133: The GEOS-Chem model needs to be described in the main text in the Methods section.
- g. Lines 207 and 211: Atypical, With respect to what? Polluted conditions?
- h. Lines 214-216: This statement is not supported by the results presented in Fig. 3.
- i. Line 248: "traditional view involving continuous oxidative decay of NO_x...". Not accurate. The traditional view includes regeneration of NO_x from HNO₃, HNO₄ and organic nitrates.
- j. Fig. 4: Please add GEOS-Chem results to the figure, too.
- k. Fig. 5: What does the dashed red line denote?

Reviewer #2 (Remarks to the Author):

Review of "External Cycling of NO_x promotes the Oxidative Capacity of the Atmosphere" by Ye et al.

This paper presents an integrated view of the atmospheric chemical cycling of reactive and reservoir nitrogen species, drawing upon analyses of a collection of airborne atmospheric composition measurements. Cycling / recycling of NO_x species such as NO and NO₂ controls many atmospheric chemical processes, ranging from oxidative capacity (the abundance of OH, production rate of ozone) to formation and nature of secondary inorganic aerosol (affecting health and radiative transfer / climate). A number of recent studies, including the authors own, have presented evidence for external NO_x cycling mechanisms – whereby NO / NO₂ are regenerated from nominal reservoirs such as HNO₃, in some cases through the intermediate HONO. Here, the authors present a more comprehensive synthesis from observations across a range of chemical environments, demonstrating that "non-traditional" (i.e., not considered in most atmospheric chemical models) mechanisms (external cycling pathways) are quantitatively necessary throughout atmospheric chemistry, further evidencing the need for critical revision of understanding.

The paper in essence provides a new and more comprehensive treatment of / demonstration of the evidence for external NO_x cycling across chemical environments, considered in an integrated manner. The basic occurrence of these mechanisms has been demonstrated across specific environments previously, and published (including by the authors). The novelty here is the integrated systematic analysis across the environmental (chemical condition – specifically NO_x abundance) parameter space.

It might be helpful for the authors to comment on the extent to which new data are presented here, vs that which has been used previously (eg as per comments around L102/L103)

Comments:

-The paper would benefit from a much clearer explanation / definition of internal and external NO_x cycling, both in the text and in Fig 1 (consider including the established and new reactions/processes) - and in the abstract (see below).

-L81 the mechanism need not be heterogeneous (or indeed variable at all). However the flux or rate of recycling through new mechanism(s) will vary depending on the environmental conditions.

L87/88 as above the rate of external cycling will vary with the chemical conditions. Is "account for" the right term ?

-L97 Must define low NO_x/high NO_x etc.

-L106/L107 – Distinguish between rate constants and reaction flux / process importance. The former will not change but their significance will, depending upon conditions. "expected environmental variability" – what does this mean ?

-L115 – Fig 2 shows a distribution markedly different from the model (which is the point of course). How does this square with agreeing with accepted distribution – maybe need to be more quantitative here. Plotting these data in frequency space shows the distribution of conditions probed by the flights, but is subject to the airmasses probed by these. Are these distributions representative of the FT or BL overall (e.g. from GEOS-Chem) ?

-L148/149 how can turnover of HONO produce HONO as a product – maybe this is terminology / phrasing

-L160 there are a number of photosensitized NO₂ reactions possible (L167: more than one thing may be happening so the inference only applies in isolation)

-L203 isn't this (flat profiles) evident – I think has been noted in some of the previous papers presenting evidence for additional mechanisms

-L226 – GEOS-Chem certainly includes radical chemistry...

-L232-236 must include the mechanisms S1 and S2 with "reactions" in the main paper / figure

-L248 the traditional view is challenged in the prior papers also (also L280)

-L272 what does "agree on the environmental variability" mean ? If you mean "the data are consistent with the occurrence of external cycling mechanism(s) across the chemical parameter space explored" ?

-L278 don't the data indicate mechanistic consistency across environmental conditions ?

-L282 define high NO_x & contrast with the data here

-L285 is this what fig 5 shows (slow change) ?

-L310 why does that fact that (most) OH production is photochemical amplify the recycling mechanism ?

-L322 ...need for revision... *in models*

Minor Comments

Abstract is unclear esp around lines 44-46. 48 – is this really global, either geographically or in terms of chemical coordinates (NO_x abundance) ?

-L63 NO_x catalysed processes – needs explanation. This section is not clear – even for the atmospheric chemistry expert

-L99 and around check tenses

-L115 what is the "middle free troposphere"

- L141 simultaneous with what ?
- L144 Not requires – is associated with
- L172 sudden introduction of one possible interference issue – needs some background to explain why this is in the main text – may be better in SI ?
- L176 DOAS typically more accurate (not precise or sensitive) than LPAP, and with uncertainties expected to be systematic wrt abundance (relating to NO₂/HONO spectral cross-contamination in reference spectra). I don't think you can argue the DOAS is wrong vs the LPAP without much more detail.
- L198 NO_x partitioning isn't (directly) driven by JO(1D) – rather than correlation of j(NO₂) with j(O1D) – be precise ! (neglecting O1D + H₂O...)
- L199 HONO appears three times in this sentence
- L229 – Justify the power plant choice (as a perturbation to consider, not the chemical conditions of this specific one)
- Fig 3 add red/blue labels to actual figure
- Fg 4 suggest make all models red and all observations blue, in line with Figs 2 and 3
- Fig 5 define Cexternal, Cinternal – not mentioned anywhere in the manuscript
- L288-291 I don't think this technical approach detail belongs in the main paper – suggest move to SI
- L209 you mean most sensitive to small *absolute* changes in NO_x abundance
- L295 "limit thought" ?
- L306 increasing with respect to what ?

Response to reviewer's comments

Reviewer #1 (Remarks to the Author):

Ye et al. use aircraft measurements of nitrogen oxides (NO_x), nitrous acid (HONO) and the sum of oxidized reactive nitrogen species (NO_y=NO_x + reservoirs) to show that recycling of NO_x from the reservoir species is faster than what is commonly accounted for in atmospheric chemistry models and that it involves HONO as an intermediate species. The authors show that including the fast recycling of NO_x in a model improves leads to a substantial increase in NO₂ and HONO concentrations in free troposphere in the model, making it consistent with aircraft observations. This also leads to a substantial increase in the modeled OH concentration.

Previous studies have shown that recycling of NO_x from its reservoirs is faster than what is predicted by established theory, but the mechanism and rate of the recycling reactions are not clear. In this study, the authors analyze the complete set of observations from a regional aircraft campaign to provide further evidence of NO_x recycling. Previously, they had published similar results based on a smaller subset of the observations. The study uses high-quality observations, well-established models, and appropriate analysis to strengthen the evidence for NO_x recycling.

The manuscript lacks clarity, key sections are missing, and is difficult to follow. Additionally, the results, as presented, do not constitute a significant advance on this topic. However, the study can be improved by considering the following revisions:

Response: A major revision has been made taking advantage of the comments and suggestions to improve the writing clarity and better summarize our analysis strategy and the innovation contribution on this topic.

1. The manuscript is missing a discussion of the results of this work in comparison to previous studies on this topic. Specifically, how does the recycling rate compare to what has been reported in the literature and can it be explained by mechanisms that have been proposed previously? The authors themselves have published several studies on particulate nitrate photolysis as the potential mechanism, but for some reason seem reluctant to discuss this in the main text, although it is discussed in the supplement. The significance of this study depends on whether it gives an insight on these questions.

Response: Your comments truly inspire us to clarify our analysis strategy and summarize the innovation contribution on this topic. A major revision has been made accordingly. Many thanks!

Previous field studies obtained varied reaction rates of the external cycling by employment of a budget analysis for either HONO or NO₂, for example in (YE *et al.* 2016; ROMER *et al.* 2018; YE *et al.* 2018; PENG *et al.* 2022; ANDERSEN *et al.* 2023). Those varied reaction rates were usually used to vote for/against a specific external cycling mechanism, and no consensus has been reached. Alternatively, those varied reaction rates might merely represent the environmental variability of the specific external cycling mechanism or analysis uncertainty concerning the specific external cycling mechanism. In such research background, it is meaningful to step back and not to focus on any specific mechanism (at least as merely field observations are discussed), but to synthesize critical field observational evidences, summarize the fundamental characteristics, and validate the

impact of external cycling on the oxidative capacity of the atmosphere from high-NO_x to low-NO_x atmospheres.

Through such a broad lens, this manuscript takes advantage of high-quality aircraft observations and well-established models and gains meaningful insights on this topic. First, this manuscript integrated systematic analysis across high-NO_x and low-NO_x chemical coordinates to summarize two pieces of critical field evidences for the external cycling of NO_x, i.e., unexpectedly high HONO/NO₂ and NO₂/NO_y (\equiv NO_x + NO_z) ratios and atypical diurnal profiles of HONO and NO₂. Second, the external cycling features HONO as a photochemical intermediate and features increasing contribution of the external cycling to the budget of HONO and NO_x as NO_x decreases. The former feature implies an amplified perturbation of the external cycling on the oxidative capacity of the atmosphere since photochemical external cycling and photochemical production of OH overlap temporally in the daytime. The latter feature further implies an amplified perturbation of the external cycling on the oxidative capacity of the low-NO_x atmosphere since NO_x impacts OH photochemistry in a more sensitive way in low-NO_x regimes. Our data and analysis in low-NO_x regimes also better represent the general tropospheric conditions than the conclusions obtained from high-NO_x conditions in terms of chemical coordinates. Finally, external cycling largely accounted for the model underestimation of OH abundance in the low-NO_x atmosphere, confirming its impact on the oxidative capacity of the atmosphere.

Please also refer to the discussion in the abstract (Lines 39-57) and the introduction (Lines 85-110).

2. The modeling of the power plant plume (lines 226 onwards and fig. 4) is confusing. It is not clear how a “plume” would even be defined after 150-200 hours of transport and dilution with the background air. To avoid confusion, it would be clearer if only the results from the box model at steady-state were presented and compared to aircraft observations in Fig. 4.

Response: It has been rewritten in lines 265-276 as “As the GEOS-Chem does not compile detailed radical chemistry along the oxidation of VOCs (LELIEVELD *et al.* 2008; TARABORRELLI *et al.* 2012), a nearly explicit chemical model is adapted to evaluate the environmental variability of external cycling and its perturbation on OH photochemistry by simulating the photochemical evolution of a power plant plume under typical low-NO_x conditions (Methods). Measurements in a random power plant plume are chosen to initialize our model, while chemical conditions in the low-NO_x atmosphere are carefully summarized to constrain the model. The model was finally initialized with observed NO_x and HONO in the Fort Martin station power plant plume captured in RF10 and ran with typical background atmospheric conditions at noon. A conceptual photochemical evolution under solar noon conditions, rather than real-time plume diel evolution from photochemistry and dilution, was therefore simulated in our model.”

3. The analysis related to the effect of NO_x recycling on OH concentrations (Lines 306-319 and Fig. 4) is inadequate. It is based on results of modeling an emission plume that has travelled 150-200 hours, which presents certain difficulties (as mentioned in point 2). In addition, it appears that not all of the OH production and loss terms were taken into account. For instance, the OH loss rate was set to a fixed value based on the measurements of the primary OH reactants, but this probably underestimates the OH reactivity (see Thames *et al.*, 2020, 10.5194/acp-20-4013-2020).

Response: We have to admit that modeling OH is always not easy. Our strategy (please also refer to

the response to point 2) is to simulate the conceptual photochemical evolution of a power plant plume in low-NO_x conditions. The chemical model is initialized by observed NO_x and HONO in the Fort Martin station power plant plume captured in RF10 and constrained with a typical background condition at noon. The total OH loss rate was set to approximately 5 s⁻¹, which is much higher than the marine boundary layer condition as in (THAMES *et al.* 2020) or the free tropospheric condition (TRAVIS *et al.* 2020) but is representative of the terrestrial background atmosphere we mainly flew over during the field campaign. Reasonable agreement between OH modeling and OH observation in the low-NO_x atmosphere was achieved. In addition, we try to compare OH simulations with/without the external cycling included in the model to illustrate the perturbation on OH photochemistry by external cycling. Such perturbation on OH photochemistry is also comparable to a previous chemical transport model evaluation (REED *et al.* 2017; KASIBHATLA *et al.* 2018). Hence, our modeling strategy and results further support our observational conclusion about external cycling and reasonably evaluate the impact on the OH photochemical budget.

Please also refer to the discussion Line 291-316 in the text.

4. The section from lines 168 to 192 discusses the possibility of measurement artifacts for HONO, HNO₃ and particulate nitrate. Additionally, there is a known positive interference in the NO₂ chemiluminescence measurements used in this study. I recommend that this issue also be addressed within this section.

Response: This issue has also been addressed in lines 218-221 as follows: “Similarly, although reliable NO_x measurements have been reported intensively in the literature (LEE *et al.* 2009), potential positive interference is not excluded (FUCHS *et al.* 2010). However, model underestimation on such as HONO/NO₂ ratio would be even worse assuming potential positive interference for NO₂.”

5. The manuscript is lacking a crucial section - the “Methods” - which is only available in the supplement. This makes it challenging to interpret the results. Please add the “Methods” section to the main text. Specifically, the authors should provide a more detailed explanation of the box modeling approach and the different model configurations. The calculation of the C(external) source in configurations S1 and S2 is not clearly explained and needs to be addressed.

Response: The “Methods” section has been added to the main text (Lines 398-538). Calculation of the $C_{internal}$ and $C_{external}$ is explained in Lines 413-449. We adapted box modeling approach used in our previous modeling work, which is fully described in (YE *et al.* 2017). Specific model setup in this study is illustrated in Lines 469-510. Model configurations are illustrated in Table S3 and Table S4.

6. The introduction could benefit from a more detailed and specific description of the problem addressed in the manuscript, including an explanation of the relevant chemistry. Fig. 1 may not be necessary and could potentially be removed.

Response: Please refer to our response to point 1. We have introduced proxy mechanisms of external cycling and the research dilemma of being unable to conclude on the rate of any specific mechanism in our society from previous field studies. Our analysis strategy is to step back and not to focus on any specific mechanism and its kinetics (at least as merely field observations are discussed), but to synthesize critical field observational evidences, summarize the fundamental characteristics, and validate the impact of external cycling on the oxidative capacity of the atmosphere in high-NO_x and

low-NO_x atmospheres. The detailed kinetics and mechanisms of a specific external cycling mechanism are therefore not the topic of this manuscript.

7. Minor/technical comments:

a. The title of the manuscript does not accurately reflect the main findings of the study, which focus on NO_x recycling rather than the oxidative capacity of the atmosphere.

Response: The title has been rewritten as “Synthesizing evidence for the external cycling of NO_x in high-to-low-NO_x atmospheres”.

b. Line 54: hydroxyl radical

Response: It has been corrected.

c. Line 74-75: NO_x can also be recycled from alkyl nitrates, HNO₄, CH₃ONO₂, and HNO₃.

Response: These well-known reservoirs of NO_x are included in the revised manuscript (Lines 306-310).

d. Line 86: It is not obvious that a regional campaign in one season can provide a “global view”. Please clarify or consider rewording.

Response: It has been rewritten as “through broader lens” to avoid misunderstanding.

e. Lines 91-99: Measurement details need to be included in the main text in the Methods section.

Response: The “Methods” section has been added to the main text.

f. Line 133: The GEOS-Chem model needs to be described in the main text in the Methods section.

Response: The “Methods” section has been added to the main text and the description of GEOS-Chem model is included (Lines 511-534).

g. Lines 207 and 211: Atypical, With respect to what? Polluted conditions?

Response: In this context, atypical diurnal is referred to as “the lack of expected daytime minima of NO₂ and HONO, which is normally observed in a high-NO_x atmosphere.”

h. Lines 214-216: This statement is not supported by the results presented in Fig. 3.

Response: The atypical diurnals of HONO and NO₂ are evident, especially in the BL in Fig. 3.

i. Line 248: “traditional view involving continuous oxidative decay of NO_x...”. Not accurate. The traditional view includes regeneration of NO_x from HNO₃, HNO₄ and organic nitrates.

Response: It has been written as “the current chemical model scheme involving the continuous oxidative decay of NO_x and small NO_x regeneration rate” (Lines 302-303).

j. Fig. 4: Please add GEOS-Chem results to the figure, too.

Response: It has been added to the figure.

k. Fig. 5: What does the dashed red line denote?

Response: It has been deleted.

References

- Andersen, S. T., L. J. Carpenter, C. Reed, J. D. Lee, R. Chance *et al.*, 2023 Extensive field evidence for the release of HONO from the photolysis of nitrate aerosols. *Science Advances* 9.
- Fuchs, H., S. M. Ball, B. Bohn, T. Brauers, R. C. Cohen *et al.*, 2010 Intercomparison of measurements of NO₂ concentrations in the atmosphere simulation chamber SAPHIR during the NO₃Comp campaign. *Atmospheric Measurement Techniques* 3: 21-37.
- Kasibhatla, P., T. Sherwen, M. J. Evans, L. J. Carpenter, C. Reed *et al.*, 2018 Global impact of nitrate photolysis in sea-salt aerosol on NO_x, OH, and O₃ in the marine boundary layer. *Atmospheric Chemistry and Physics* 18: 11185-11203.
- Lee, J. D., S. J. Moller, K. A. Read, A. C. Lewis, L. Mendes *et al.*, 2009 Year-round measurements of nitrogen oxides and ozone in the tropical North Atlantic marine boundary layer. *Journal of Geophysical Research-Atmospheres* 114.
- Lelieveld, J., T. M. Butler, J. N. Crowley, T. J. Dillon, H. Fischer *et al.*, 2008 Atmospheric oxidation capacity sustained by a tropical forest. *Nature* 452: 737-740.
- Peng, Q. Y., B. B. Palm, C. D. Fredrickson, B. Lee, S. R. Hall *et al.*, 2022 Direct Constraints on Secondary HONO Production in Aged Wildfire Smoke From Airborne Measurements Over the Western US. *Geophysical Research Letters* 49.
- Reed, C., M. J. Evans, L. R. Crilley, W. J. Bloss, T. Sherwen *et al.*, 2017 Evidence for renoxification in the tropical marine boundary layer. *Atmospheric Chemistry and Physics* 17: 4081-4092.
- Romer, P. S., P. J. Wooldridge, J. D. Crouse, M. J. Kim, P. O. Wennberg *et al.*, 2018 Constraints on Aerosol Nitrate Photolysis as a Potential Source of HONO and NO_x. *Environmental Science & Technology* 52: 13738-13746.
- Taraborrelli, D., M. G. Lawrence, J. N. Crowley, T. J. Dillon, S. Gromov *et al.*, 2012 Hydroxyl radical buffered by isoprene oxidation over tropical forests (vol 5, pg 190, 2012). *Nature Geoscience* 5: 300-300.
- Thames, A. B., W. H. Brune, D. O. Miller, H. M. Allen, E. C. Apel *et al.*, 2020 Missing OH reactivity in the global marine boundary layer. *Atmospheric Chemistry and Physics* 20: 4013-4029.
- Travis, K. R., C. L. Heald, H. M. Allen, E. C. Apel, S. R. Arnold *et al.*, 2020 Constraining remote oxidation capacity with ATom observations. *Atmospheric Chemistry and Physics* 20: 7753-7781.
- Ye, C. X., D. E. Heard and L. K. Whalley, 2017 Evaluation of Novel Routes for NO_x Formation in Remote Regions. *Environmental Science & Technology* 51: 7442-7449.
- Ye, C. X., X. L. Zhou, D. Pu, J. Stutz, J. Festa *et al.*, 2016 Rapid cycling of reactive nitrogen in the marine boundary layer. *Nature* 532: 489-491.
- Ye, C. X., X. L. Zhou, D. Pu, J. Stutz, J. Festa *et al.*, 2018 Tropospheric HONO distribution and chemistry in the southeastern US. *Atmospheric Chemistry and Physics* 18: 9107-9120.

Reviewer #2 (Remarks to the Author):

Review of “External Cycling of NO_x promotes the Oxidative Capacity of the Atmosphere” by Ye et al.

This paper presents an integrated view of the atmospheric chemical cycling of reactive and reservoir nitrogen species, drawing upon analyses of a collection of airborne atmospheric composition measurements. Cycling / recycling of NO_x species such as NO and NO₂ controls many atmospheric chemical processes, ranging from oxidative capacity (the abundance of OH, production rate of ozone) to formation and nature of secondary inorganic aerosol (affecting health and radiative transfer / climate). A number of recent studies, including the authors own, have presented evidence for external NO_x cycling mechanisms – whereby NO / NO₂ are regenerated from nominal reservoirs such as HNO₃, in some cases through the intermediate HONO. Here, the authors present a more comprehensive synthesis from observations across a range of chemical environments, demonstrating that “non-traditional” (i.e., not considered in most atmospheric chemical models) mechanisms (external cycling pathways) are quantitatively necessary throughout atmospheric chemistry, further evidencing the need for critical revision of understanding.

The paper in essence provides a new and more comprehensive treatment of / demonstration of the evidence for external NO_x cycling across chemical environments, considered in an integrated manner. The basic occurrence of these mechanisms has been demonstrated across specific environments previously, and published (including by the authors). The novelty here is the integrated systematic analysis across the environmental (chemical condition – specifically NO_x abundance) parameter space.

It might be helpful for the authors to comment on the extent to which new data are presented here, vs that which has been used previously (eg as per comments around L102/L103)

Response: The review’s evaluation on our manuscript is enlightening. We revised our manuscript to summarize our analysis strategy and the innovation contribution in a more accurate way that the reviewer has helped to summarize here. We deeply appreciate the help. The suggestion to comment on the new data is also followed (Lines 124-127).

Comments:

-The paper would benefit from a much clearer explanation / definition of internal and external NO_x cycling, both in the text and in Fig 1 (consider including the established and new reactions/processes) - and in the abstract (see below).

Response: The suggestions are accepted.

-L81 the mechanism need not be heterogeneous (or indeed variable at all). However, the flux or rate of recycling through new mechanism(s) will vary depending on the environmental conditions.

Response: The sentence has been rewritten in lines 95-99 as “However, the heterogeneous or multiphase nature of this reaction highlights its substantial environmental variability in both the kinetics and mechanisms, which does not allow us to simply extrapolate results from laboratory studies to field studies or compare results among field studies in various chemical regimes”

L87/88 as above the rate of external cycling will vary with the chemical conditions. Is “account for” the right term ?

Response: The sentence has been rewritten in lines 102-109 as “To solve this dilemma, we suggest synthesizing critical observational and model evidence, summarizing the fundamental characteristics, and validating the impact of external cycling on the oxidative capacity of the atmosphere in high- to low-NO_x atmospheres through broader lens, rather than attempting to establish the kinetics and mechanism for a specific mechanism of external cycling.”

-L97 Must define low NO_x/high NO_x etc.

Response: The definition of low-NO_x environments has been added (Line 119). Based on our observation, below 500 pptv appears to be good enough definition of low-NO_x atmosphere in the context.

-L106/L107 – Distinguish between rate constants and reaction flux / process importance. The former will not change but their significance will, depending upon conditions. “expected environmental variability” – what does this mean ?

Response: The sentence has been rewritten as “The reaction rate constant of the external cycling implied from the missing HONO source among these reports deviates by more than one order of magnitude, leading to conclusions on the environmental variability of the external cycling, or more likely different opinions on external cycling mechanisms.” (Lines 129-133)

-L115 – Fig 2 shows a distribution markedly different from the model (which is the point of course). How does this square with agreeing with accepted distribution – maybe need to be more quantitative here. Plotting these data in frequency space shows the distribution of conditions probed by the flights, but is subject to the airmasses probed by these. Are these distributions representative of the FT or BL overall (e.g. from GEOS-Chem) ?

Response: We have reviewed the distributions observed in other remote boundary layer and free troposphere by other aircraft measurements, for example in remote Atlantic (ANDERSEN *et al.* 2023), remote forested region (ZHANG *et al.* 2009). Our distributions are comparable with those measured in previous research, with NO_x distributing at hundreds pptv and HONO at tens pptv, suggesting that the aircraft measurements well represent the low-NO_x atmosphere.

-L148/149 how can turnover of HONO produce HONO as a product – maybe this is terminology / phrasing

Response: In the HONO-NO_x internal cycling, the turnover rate of NO_x to produce HONO should balance the turnover rate of HONO to produce NO_x.

-L160 there are a number of photosensitized NO₂ reactions possible (L167: more than one thing may be happening so the inference only applies in isolation)

Response: Yes, there are various photosensitized reactions of NO₂ with NO₂ concentration, and the

photosensitization reaction surface (i.e., organic aerosol) is the major variable in the reaction. Even with the upper limit evaluation of the photosensitization reaction of NO₂, the observed HONO budget and high HONO/NO₂ cannot be accounted for.

-L203 isn't this (flat profiles) evident – I think has been noted in some of the previous papers presenting evidence for additional mechanisms

Response: Herein, we refer to the bell-shaped diel pattern and the higher-than-expected noontime concentration of HONO or NO₂ observed in the background atmosphere as “atypical diel” of HONO and NO₂ relative to the typical U-shaped diel observed in the high-NO_x atmosphere. Atypical diels include flat ones and bell-shaped ones, probably depending on the strength of the external cycling. Such atypical diels, as a matter of fact, might be quite popular in the low-NO_x background atmosphere. Previous papers summarize such atypical diurnal profiles in the clean marine boundary layer and snow-covered polar areas. In addition to previous papers, we find such atypical diurnal profiles in the low-NO_x terrestrial boundary layer and the free troposphere. We have revised the paragraph to clarify these two points (Lines 226-240).

-L226 – GEOS-Chem certainly includes radical chemistry...

Response: The sentence has been rewritten as “As GEOS-Chem does not compile detailed radical chemistry along the oxidation mechanism of VOCs (LELIEVELD *et al.* 2008; TARABORRELLI *et al.* 2012)”

-L232-236 must include the mechanisms S1 and S2 with “reactions” in the main paper / figure

Response: It has been included in the revised manuscript (Lines 489-490).

-L248 the traditional view is challenged in the prior papers also (also L280)

Response: It has been rephrased as “current chemical model scheme” (Line 302).

-L272 what does “agree on the environmental variability” mean ? If you mean “the data are consistent with the occurrence of external cycling mechanism(s) across the chemical parameter space explored” ?

Response: The environmental variability pattern of the external cycling is “increasing perturbation by external cycling on atmospheric budget of NO_x and HONO as NO_x decreases”. “Agree on” has been rephrased as “consistently reveal”. (Line 331)

-L278 don't the data indicate mechanistic consistency across environmental conditions ?

Response: The data do indicate mechanistic consistency across environmental conditions. However, variability in the rate of the external cycling and its perturbation on the chemical budget of HONO and NO₂ are also expected. The environmental variability pattern mainly originates from the accumulation of NO_z (the external cycling precursor) as NO_x decreases. The environmental variability of the rate constant of external cycling, i.e., the high rate constant of pNO₃ photolysis in a low NO_x atmosphere, might also contribute (YE *et al.* 2017; ANDERSEN *et al.* 2023), but it is not considered in the chemical model.

-L282 define high NO_x & contrast with the data here

Response: Based on our observation, below 500 pptv appears to be good enough definition of low-NO_x atmosphere, which above 500 pptv is high -NO_x atmosphere. It has been defined in the revised manuscript.

-L285 is this what fig 5 shows (slow change)?

Response: It has been rephrased as “The shifting of the HONO/NO₂ ratio as NO_x decreases in the high-NO_x atmosphere is so slow in a so narrow range of typically from 0.01 to 0.1 that it might have been mistaken to be unchanging (Fig. 4-5).”

-L310 why does that fact that (most) OH production is photochemical amplify the recycling mechanism ?

Response: The perturbation of the external cycling on OH production depends on two features of the external cycling, i.e., the photochemical nature and increasing contribution to the budget of NO_x as NO_x decreases. This point has been clarified in the revised manuscript (Lines 331-366).

-L322 ...need for revision... *in models*

Response: This correction has been accepted.

Minor Comments

Abstract is unclear esp around lines 44-46. 48 – is this really global, either geographically or in terms of chemical coordinates (NO_x abundance) ?

Response: It has been rephrased as “broader lens to view external cycling and its perturbations on the atmospheric oxidative capacity”. (Lines 53-54)

-L63 NO_x catalysed processes – needs explanation. This section is not clear – even for the atmospheric chemistry expert

Response: The paragraph has been rewritten. (Lines 63-65)

-L99 and around check tenses

Response: It has been revised.

-L115 what is the “middle free troposphere”

Response: “middle” has been deleted.

-L141 simultaneous with what ?

Response: It has been rephrased as “moreover”.

-L144 Not requires – is associated with

Response: It has been revised.

-L172 sudden introduction of one possible interference issue – needs some background to explain why this is in the main text – may be better in SI ?

Response: Measurement interferences are challenged by other reviewers as an alternative reason for the model-observation discrepancies. This section is to carefully examine this alternative reason.

-L176 DOAS typically more accurate (not precise or sensitive) than LPAP, and with uncertainties expected to be systematic wrt abundance (relating to NO₂/HONO spectral cross-contamination in reference spectra). I don't think you can argue the DOAS is wrong vs the LPAP without much more detail.

Response: Fair enough. It has been revised as "DOAS shows smaller value relative to LAPA in the lower concentration range". (Line 204)

-L198 NO_x partitioning isn't (directly) driven by JO(1D) – rather than correlation of j(NO₂) with j(O¹D) – be precise ! (neglecting O¹D + H₂O...)

Response: It has been revised.

-L199 HONO appears three times in this sentence

Response: It has been revised.

-L229 – Justify the power plant choice (as a perturbation to consider, not the chemical conditions of this specific one)

Response: This is a nice suggestion. This point has been clarified. (Lines 265-276)

-Fig 3 add red/blue labels to actual figure

Response: This has been stated in the figure caption. Red/blue labels make the figure quite busy.

-Fig 4 suggest make all models red and all observations blue, in line with Figs 2 and 3

Response: This is the way we plot Figs. 2 and 3.

-Fig 5 define Cexternal, Cinternal – not mentioned anywhere in the manuscript

Response: They have been defined at their first appearance. (Lines 335 & Line 348)

-L288-291 I don't think this technical approach detail belongs in the main paper – suggest move to SI

Response: Other reviewer suggested to incorporate measurement interference and uncertainties into the analysis.

-L209 you mean most sensitive to small *absolute* changes in NO_x abundance

Response: Yes! This has been clarified.

-L295 "limit thought" ?

Response: It has been revised to "An assumed extreme case".

-L306 increasing with respect to what ?

Response: It has been revised as "Both the photochemical nature and the increasing contribution of the external cycling to the budget of HONO and NO_x as NO_x decreases in the low-NO_x atmosphere

might have amplified its perturbation on OH photochemistry in the troposphere”. (Lines 377-379)

References

- Andersen, S. T., L. J. Carpenter, C. Reed, J. D. Lee, R. Chance *et al.*, 2023 Extensive field evidence for the release of HONO from the photolysis of nitrate aerosols. *Science Advances* 9.
- Lelieveld, J., T. M. Butler, J. N. Crowley, T. J. Dillon, H. Fischer *et al.*, 2008 Atmospheric oxidation capacity sustained by a tropical forest. *Nature* 452: 737-740.
- Taraborrelli, D., M. G. Lawrence, J. N. Crowley, T. J. Dillon, S. Gromov *et al.*, 2012 Hydroxyl radical buffered by isoprene oxidation over tropical forests (vol 5, pg 190, 2012). *Nature Geoscience* 5: 300-300.
- Ye, C. X., D. E. Heard and L. K. Whalley, 2017 Evaluation of Novel Routes for NO_x Formation in Remote Regions. *Environmental Science & Technology* 51: 7442-7449.
- Zhang, N., X. L. Zhou, P. B. Shepson, H. L. Gao, M. Alaghmand *et al.*, 2009 Aircraft measurement of HONO vertical profiles over a forested region. *Geophysical Research Letters* 36.

Reviewer #1 (Remarks to the Author):

The authors' response addresses some of my comments on the original submission. However, a discussion of how the results of this work relate to previous work on this topic in terms of the rates and mechanisms is still not included, although the authors have added some introductory text. Again, without this, I do not see this work as providing significant new insights.

I also don't understand why there is a reluctance to discuss mechanisms, even though the study uses particulate nitrate photolysis as the "external proxy mechanism" (Lines 471-473 in the revision). If the intent is "to step back and not to focus on any specific mechanism," it is not clear to me why to begin by assuming that the recycling will have characteristics of particulate nitrate photolysis, that is, driven by the UV flux, dependent on particulate nitrate concentrations, and produce either HONO or NO₂.

The section on modeling the power plant plume in low-NO_x conditions is still unclear.

Reviewer #3 (Remarks to the Author):

[Reviewer 2]

The authors have responded well to almost all the specific points I flagged in my initial review, and the manuscript is much improved, with all the science points addressed.

I'm still troubled by how hard it is to follow the argument in the manuscript though – this is an issue of phrasing and precision, not science. Can I suggest the authors seek a "fresh pair of eyes" to look at the text throughout, but in particular the Introduction (L60-110) and final conclusions L377-L396 and really examine what each paragraph is trying to convey, and what precisely each sentence actually says.

Phrasing Comments :

-Introduction to / explanation of Fig 1 is lacking. Must include a really good concise definition of both internal and external cycling – this is critical – as the key terms used throughout the paper

L92 the *effective* rate constant ? Needs some more words somewhere in the sentence

L94 I don't think a rate constant can vote. I know what you mean but be precise in the phrasing – what you mean is "the range of values for rate constants / fundamental parameters which describe the individual candidate mechanisms differ widely between studies, precluding (as yet) confident confirmation of one or more dominant mechanisms"

L97 – the parameters/rate constants don't vary with the chemical environment – if they are truly fundamental properties of a given reaction/mechanism – but the rate of the mechanism/reaction overall will. Again need to be much sharper between *parameters/rate constants* and reaction/process *rates*. Also L342

L101 – tighten up what you mean by environmental variability. If fundamental parameters are obtained from the various literature studies – these should be independent of the NO_x concentration (or H₂O, j etc). If they vary with the chemical environment – then the mechanism is wrong/incomplete.

L131 – leading to conclusions – what are they – make this clearer. Suggest avoid words like "opinions" – the point is that different mechanisms (or parameters/rate constants) lead to different predicted NO_x/HONO/OH variation in the atmosphere, suggesting that the dominant mechanism(s) have yet to be identified (or correctly characterized) ?

L252 – neither of the competing mechanism – interference is not a mechanism / you only mention one mechanism

L265 – needs a new sentence to explain (big picture) why you are looking at the plume experiment. Start further “back” in the logic than the GEOS-Chem chemical scheme – what is the big picture aim/point of this ? “To explore the variation in NO_x, HONO and OH with airmass age or extent of chemical processing, we examine one representative case – the evolution of chemical composition in a notional power plant plume, which covers a wider range of chemical conditions / NO_x space, and relates <how ?> to the observation parameter space”

L357-L362 I still think these specific technical details belong in the SI – they read very incongruously here

High / Low NO_x: from an atmospheric chemistry perspective these are often considered in terms of the fate of peroxy radicals HO₂/RO₂ (ie do they terminate together or propagate with NO) – which would put “low NO_x” way below 500 ppt. ie 500 pptv is going to be “high” to some. Maybe address here as simply as “here, low NO_x is taken to refer to measurements where NO_x < 500 pptv, which accounts for xx% of our observations – although we note that the outflow of the continental US will show higher NO_x levels than the global remote troposphere” or something

Response to reviewers' comments

Reviewer #1 (Remarks to the Author):

The authors' response addresses some of my comments on the original submission. However, a discussion of how the results of this work relate to previous work on this topic in terms of the rates and mechanisms is still not included, although the authors have added some introductory text. Again, without this, I do not see this work as providing significant new insights.

Response: In this manuscript, we synthesized field evidence and fundamental characteristics of the external cycling rather than any specific or dominant mechanism of the external cycling, even though a specific mechanism, i.e., the photolysis of particulate nitrate, could potentially be responsible for the discussed external cycling. Currently, only limited data are available for pNO₃ photolysis rate constants in a few atmospheric environments and with large uncertainties. It is therefore difficult to extrapolate results from laboratory studies to field studies or to directly compare results among field studies. It has also precluded as yet confident confirmation of pNO₃ photolysis or other reactions as the dominant mechanism of the external cycling or direct characterization of the external cycling by exploring any specific mechanism in the atmosphere. One natural step is to collect more data in the laboratory and field to explore the specific kinetics of a dominant external cycling mechanism. Alternatively, we synthesize field evidence, fundamental characteristics, and atmospheric impact of the external cycling through a broader lens. We explained this strategy and our contribution to this topic in the revised manuscript in lines 80-114.

In lines 80-114, "Identifying and exploring a specific mechanism is a natural step in characterizing external cycling and quantifying its impact on the oxidative capacity of the atmosphere. The external cycling pathways proposed in the literature include at least three mechanisms, i.e., surface-catalyzed photolysis of absorbed nitrate (nitrate_{abs}) on snow/ice surfaces and possibly also on aerosol/ambient surfaces^{4,20,24,27-38}, nitrification/denitrification in the soil^{16,19} and the thermal decomposition of peroxyacetyl nitrate (PAN). Among these mechanisms, surface-catalyzed photolysis of nitrate_{abs} on aerosol surfaces, referred to as pNO₃ photolysis, occurs in situ in the air column and therefore potentially perturbs the distribution of NO_x and hence the oxidative capacity of the atmosphere from the lower troposphere to the upper troposphere³⁵. pNO₃ photolysis has been intensively explored in laboratory and field studies^{4,18,21-23,34,36,37,39}. The reaction rate constant of pNO₃ photolysis varies over at least two orders of magnitude in high-to-low-NO_x atmospheres by employing a budget analysis for either HONO or NO₂ and by assuming that pNO₃ photolysis fully accounts for the missing source of HONO or NO₂ in the field^{4,21,32,34,38}. Laboratory studies on a variety of pNO₃ samples have confirmed that pNO₃ photolysis is greatly enhanced compared to that of gaseous HNO₃ and that the pNO₃ photolysis rate constant is highly variable, over 3 orders of magnitude^{30,36,37}. Based on these laboratory and field studies, Ye et al. and Andersen et al. have also revealed that pNO₃ photolysis is surface-catalyzed in nature and is greatly affected by the physicochemical properties of aerosol particles, such as pNO₃ loading, chemical composition and particle size^{30,38}. Efforts in characterizing atmospheric aerosol properties and their photochemical reactivities are critical in quantitatively understanding the role of pNO₃ photolysis in the external cycling. However, the limited availability of the pNO₃ photolysis rate constant in only a few atmospheric environments, its large variability, and potentially large uncertainties make it difficult to extrapolate results from laboratory studies to field studies or directly compare results among field

studies^{18,21,22,30,33,34,36-39}. The variability and potentially large uncertainties in the rate constant have also precluded as yet confident confirmation of pNO₃ photolysis or other reactions as the dominant mechanism of the external cycling or direct characterization of the external cycling in the atmosphere. To solve this dilemma, we suggest synthesizing critical observational and model evidence, summarizing the fundamental characteristics, and quantifying the impact of the external cycling on the oxidative capacity of the atmosphere in high- to low-NO_x atmospheres, through a broader lens, rather than attempting to establish the kinetics or the dominant mechanism of the external cycling.”

I also don't understand why there is a reluctance to discuss mechanisms, even though the study uses particulate nitrate photolysis as the "external proxy mechanism" (Lines 471-473 in the revision). If the intent is "to step back and not to focus on any specific mechanism," it is not clear to me why to begin by assuming that the recycling will have characteristics of particulate nitrate photolysis, that is, driven by the UV flux, dependent on particulate nitrate concentrations, and produce either HONO or NO₂.

Response: Please also see the response to the previous question.

We have discussed the photolysis of particulate nitrate as a major daytime HONO source in our earlier papers (Ye et al., *Nature*, 2016). We also tried to provide the information relevant to this mechanism in the introduction section. However, a very wide range of pNO₃ photolysis rate constants from relatively few measurements makes the extrapolation difficult. Instead, we chose to examine the external cycling in more general terms using the comprehensive dataset obtained from an aircraft measurement campaign. Our external cycling argument is based on our field observations and deductions that HONO was an intermediate in the external cycling of NO_x, and that the external cycling compensates for the daytime oxidation loss of NO₂ and HONO and that external cycling contributed more to the budget of HONO and NO_x in low-NO_x atmospheres. Please see the related discussion in the revised manuscript.

In lines 173-182, we discuss HONO as an intermediate based on field observations, “Moreover, model underestimation of the HONO/NO₂ ratio and HONO/NO_y further revealed that HONO might be an intermediate in the external cycling of NO_x, as HONO regenerated in the external cycling would rapidly photolyse to produce NO_x (Fig. 2). It was speculated that the relatively slow turnover rate of NO_x to produce HONO was associated with a low HONO/NO₂ ratio (<0.05, as commonly observed in high-NO_x atmospheres⁹) to balance the rapid turnover rate of HONO to produce NO_x via HONO photolysis and the Leighton cycle. Herein, a much higher HONO/NO₂ ratio relative to the PSS prediction in GEOS-Chem indicated a net turnover of HONO to produce NO_x in the external cycling and presented HONO as an intermediate product.”

In lines 249-257, we summarized the role of external cycling in compensate the daytime loss of HONO and NO₂, “However, we observed atypical diurnal profiles that lacked daytime minima of NO₂ and HONO in more general BL, adding to the previous observation of the same kind in the clean marine boundary layer and polar boundary layer and first in the FT (Fig. 3). This observation also differed from the GEOS-Chem simulation which excluded external cycling and simulated the expected diurnal profiles of NO₂ and HONO, especially in the BL (Fig. 3). Therefore, the atypical diurnal profiles of NO₂ and HONO provided further observational evidence of the temporal distribution of reactive nitrogen to verify the external cycling of NO_x and HONO.”

The section on modeling the power plant plume in low-NO_x conditions is still unclear.

Response: Following the advice of other referees, we have added new sentences and explained why and how we construct the model in lines 271-299. The model setup and initialization conditions are briefly described in the Methods section (lines 480-500) and summarized in Table S4.

Lines 270-298, “As the GEOS-Chem did not compile detailed radical chemistry along the oxidation mechanism of VOCs^{51,52}, a nearly-explicit chemical model (MCM version 3.3.1, <http://mcm.leeds.ac.uk/MCM/>) was adapted to simulate the photochemical evolution of a random power plant plume (Fort Martin station power plant plume captured in RF10, Methods) to represent nitrogen cycling photochemistry across various NO_x regimes over a large geographic area. The objectives of the chemical model simulation were to explore the fundamental characteristics of external cycling (i.e., HONO being an intermediate, determinative role of external cycling in the observed high ratio of HONO/NO₂ and NO₂/NO_y, and the perturbation of external cycling on OH photochemistry) along with the aging of the plume. To be more specific, measurements in the power plant plume were chosen to initialize our model, while chemical conditions in the low-NO_x atmosphere were carefully summarized to constrain the model. To focus on the chemical evolution of composition in the plume, a conceptual photochemical evolution under solar noon conditions was simulated in our model. To avoid discussion on any specific external cycling mechanism and related arguments on the reaction rate of specific external cycling routes, a proxy mechanism for the external cycling employing pNO₃ as a representative of NO_z species and the precursor of HONO and NO_x in the external cycling was included in the chemical model scheme. The pNO₃ photolysis rate or the reaction rate of a general NO_z species was set up based on the assumption that the external cycling could fully account for the unknown source of HONO or NO₂. Previous field observations or laboratory measurements of the photolysis rate constant were not referred to or compared with, as only a proxy mechanism, rather than a specific mechanism, of the external cycling was the very core of the discussion. Three independent models were run: one excluded the external cycling of NO_x and HONO (model S0), and the other two included external cycling, with/without HONO as a NO_x intermediate (model S1-S2). Since the external cycling was verified and HONO was identified as an intermediate product based on our observations, chemical model S1 was expected to best represent our observations on the distribution of HONO and NO₂ in varied NO_x regimes.”

Reference

Ye, C. X., X. L. Zhou, D. Pu, J. Stutz, J. Festa et al., 2016 Rapid cycling of reactive nitrogen in the marine boundary layer. *Nature* 532: 489-491.

Reviewer #3 (Remarks to the Author):

[Reviewer 2]

The authors have responded well to almost all the specific points I flagged in my initial review, and the manuscript is much improved, with all the science points addressed.

I'm still troubled by how hard it is to follow the argument in the manuscript though – this is an issue of phrasing and precision, not science. Can I suggest the authors seek a “fresh pair of eyes” to look at the text throughout, but in particular the Introduction (L60-110) and final conclusions L377-L396 and really examine what each paragraph is trying to convey, and what precisely each sentence actually says.

Response: We deeply appreciate the reviewer's encouragement and help! We have revised our manuscript accordingly.

Phrasing Comments :

-Introduction to / explanation of Fig 1 is lacking. Must include a really good concise definition of both internal and external cycling – this is critical – as the key terms used throughout the paper

Response: Our revision is shown in lines 57-75 as “HONO and NO_x are closely coupled in their NO_x-HONO internal cycling, referred to as internal cycling in this context (Fig. 1). Specifically, heterogeneous reactions of NO₂ on ambient surfaces and gas-phase reactions between NO and OH have been intensively examined as major HONO formation routes in the internal cycle⁷⁻¹⁵. Since HONO photolyzes much faster, turnover routes of NO_x to recycle HONO are the rate-limiting steps of the internal cycling. OH is a net product in the internal cycling even if only HONO production via heterogeneous reactions of NO₂ is considered. Any external source (or sink) of HONO or NO_x would promote (or suppress) the internal cycling. Apart from the internal cycling, NO_x ages to form its more oxidized reservoirs, which are referred to as NO_z, as airmasses are transported away from source regions. NO_x aging processes suppress the internal cycling and diminish the role of HONO photolysis in OH budget. However, model underestimation of the NO_x/NO_z ratio observations suggests unknown or underappreciated NO_x regeneration pathways from NO_z in low-NO_x atmospheres^{4,16-28}. NO_x regeneration pathways, in contrast to the internal cycling, produce “new” NO_x and hence are designated as the external cycling of NO_x (Fig. 1). External cycling naturally promotes the internal cycling via at least the external source of NO_x. Consequently, secondary OH production via NO_x regeneration, and net primary OH production via HONO regeneration could greatly perturb OH photochemical budget.”

L92 the *effective* rate constant ? Needs some more words somewhere in the sentence

Response: The sentence has been rewritten as “The reaction rate constant of pNO₃ photolysis varies over at least two orders of magnitude in high-to-low-NO_x atmospheres by employing a budget analysis for either HONO or NO₂ and by assuming that pNO₃ photolysis fully accounts for the missing source of HONO or NO₂ in the field^{4,21,32,34,38}.” (Lines 91-94)

L94 I don't think a rate constant can vary. I know what you mean but be precise in the phrasing – what you mean is “the range of values for rate constants / fundamental parameters which

describe the individual candidate mechanisms differ widely between studies, precluding (as yet) confident confirmation of one or more dominant mechanisms”

Response: The suggested phrasing is more precise. Our revision shown in lines 94-109 as “Laboratory studies on a variety of pNO₃ samples have confirmed that pNO₃ photolysis is greatly enhanced compared to that of gaseous HNO₃ and that the pNO₃ photolysis rate constant is highly variable, over 3 orders of magnitude^{30,36,37} Based on these laboratory and field studies, Ye et al. and Andersen et al. have also revealed that pNO₃ photolysis is surface-catalyzed in nature and is greatly affected by the physicochemical properties of aerosol particles, such as pNO₃ loading, chemical composition and particle size^{30,38}. Efforts in characterizing atmospheric aerosol properties and their photochemical reactivities are critical in quantitatively understanding the role of pNO₃ photolysis in the external cycling. However, the limited availability of the pNO₃ photolysis rate constant in only a few atmospheric environments, its large variability, and potentially large uncertainties make it difficult to extrapolate results from laboratory studies to field studies or directly compare results among field studies^{18,21,22,30,33,34,36-39}. The variability and potentially large uncertainties in the rate constant have also precluded as yet confident confirmation of pNO₃ photolysis or other reactions as the dominant mechanism of the external cycling or direct characterization of the external cycling in the atmosphere.”

L97 - the parameters/rate constants don't vary with the chemical environment - if they are truly fundamental properties of a given reaction/mechanism - but the rate of the mechanism/reaction overall will. Again need to be much sharper between *parameters/rate constants* and reaction/process *rates*. Also L342

Response: We have rephrased it. Please also refer to the last response.

L101 - tighten up what you mean by environmental variability. If fundamental parameters are obtained from the various literature studies - these should be independent of the NO_x concentration (or H₂O, j etc). If they vary with the chemical environment - then the mechanism is wrong/incomplete.

Response: We have rephrased it as shown in the above response.

L131 - leading to conclusions - what are they - make this clearer. Suggest avoid words like “opinions” - the point is that different mechanisms (or parameters/rate constants) lead to different predicted NO_x/HONO/OH variation in the atmosphere, suggesting that the dominant mechanism(s) have yet to be identified (or correctly characterized) ?

Response: We have rephrased it in Lines 137-140 as “The reaction rate constant of the specific external cycling route implied from the missing HONO source among these reports deviated by more than two orders of magnitude, reaching no consensus in the dominant external cycling route or atmospheric variability in the external cycling^{4,21,32,34,38}.”

L252 - neither of the competing mechanism - interference is not a mechanism / you only mention one mechanism

Response: We have rephrased it in Lines 257-260 as “Notably, neither of the internal cycling mechanisms, such as the photosensitization reaction of NO₂, nor potential measurement interferences of HONO or NO₂, were able to reconcile model–observation discrepancies in the

diurnal profiles of NO₂ and HONO.

L265 - needs a new sentence to explain (big picture) why you are looking at the plume experiment. Start further “back” in the logic than the GEOS-Chem chemical scheme - what is the big picture aim/point of this? “To explore the variation in NO_x, HONO and OH with air mass age or extent of chemical processing, we examine one representative case - the evolution of chemical composition in a notional power plant plume, which covers a wider range of chemical conditions / NO_x space, and relates <how ?> to the observation parameter space”

Response: We have rephrased the sentence in Lines 270-279 as “As the GEOS-Chem did not compile detailed radical chemistry along the oxidation mechanism of VOCs^{51,52}, a nearly-explicit chemical model (MCM version 3.3.1, <http://mcm.leeds.ac.uk/MCM/>) was adapted to simulate the photochemical evolution of a random power plant plume (Fort Martin station power plant plume captured in RF10, Methods) to represent nitrogen cycling photochemistry across various NO_x regimes over a large geographic area. The objectives of the chemical model simulation were to explore the fundamental characteristics of external cycling (i.e., HONO being an intermediate, determinative role of external cycling in the observed high ratio of HONO/NO₂ and NO₂/NO_y, and the perturbation of external cycling on OH photochemistry) along with the aging of the plume.”

L357-L362 I still think these specific technical details belong in the SI - they read very incongruously here

Response: These technical details explain why NO_y might be underestimated in our measurement, and they are removed to the Methods Section (Lines 498-506).

High / Low NO_x: from an atmospheric chemistry perspective these are often considered in terms of the fate of peroxy radicals HO₂/RO₂ (ie do they terminate together or propagate with NO) - which would put “low NO_x” way below 500 ppt. ie 500 pptv is going to be “high” to some. Maybe address here as simply as “here, low NO_x is taken to refer to measurements where NO_x < 500 pptv, which accounts for xx% of our observations - although we note that the outflow of the continental US will show higher NO_x levels than the global remote troposphere” or something

Response: We have rephrased the sentence in lines 124-128 as “Herein, we defined low-NO_x and high-NO_x regimes with a NO_x concentration threshold of 500 pptv, given that 500 pptv represented the upper limit concentration of NO_x in the remote troposphere. Furthermore, 500 pptv appeared to be a turning point for the external cycling to be key sources of HONO and NO_x (see below).